# Zero-dimensional contrail models could underpredict lifetime optical depth

Caleb Akhtar Martínez<sup>1</sup>, Sebastian D. Eastham<sup>2</sup>, Jerome P. Jarrett<sup>1</sup>

- <sup>1</sup>Department of Engineering, University of Cambridge, Cambridge, CB2 1PZ, United Kingdom
- <sup>2</sup>Department of Aeronautics, Imperial College London, London, SW7 2AZ, United Kingdom

Correspondence to: Caleb Akhtar Martínez (ca525@cam.ac.uk)

**Abstract.** Proposed contrail avoidance schemes rely on being able to robustly predict which contrails cause the most climate warming. However, it has not yet been shown that different contrail models agree sufficiently to support the targeting of individual contrails by climate impact. To address this, we compare the most widespread contrail model, CoCiP, to a higherfidelity contrail model, APCEMM, under parametrized meteorological conditions. The results show that the time-integrated total extinction (a proxy for climate impact) in APCEMM is 3.8 times that in CoCiP, and that the models have opposite sensitivities of their time-integrated total extinction to relative humidity. We argue that these differences are due to the differing representations of the distribution of ice particles in space and in size across the contrails. The use of a monodisperse ice particle size distribution in a Gaussian plume means that CoCiP models the contrail exclusively as an accelerating, falling mass. The use of a spatially gridded and size-resolved aerosol scheme allows APCEMM to represent the separation of the precipitation plume from the contrail core, hence modelling behavior beyond the initial phase in which the contrail grows unconstrained. This behavior is consistent with prior large eddy simulation studies, and it accounts for 92 % of the aggregate APCEMM time-integrated total extinction across all simulations. This suggests that models lacking a sizeresolved aerosol scheme may underestimate the time-integrated total extinction. While a strategy avoiding a given proportion of persistent contrails in an unbiased way is still expected to yield a proportional reduction in the time-integrated total extinction, implementing strategies using contrail models to select the specific contrails to avoid may lead to fewer reductions in the time-integrated total extinction, primarily due to the current level of disagreement between models. We therefore recommend more research to establish confidence in model predictions at later contrail ages.

#### 1 Introduction

- It is estimated that aviation generates 3.5 % of all anthropogenic effective radiative forcing (ERF) and that contrail cirrus contributes 67% more ERF than the carbon dioxide produced by aircraft (Lee et al., 2021). These estimates indicate that strategies aiming to reduce contrail cirrus could provide benefits of a similar magnitude as the elimination of aviation-induced CO<sub>2</sub> emissions. The simplest strategy involves attempting to avoid all contrails, but schemes targeting only the most warming contrails (Teoh et al. 2020a; Teoh et al. 2020b) are also under consideration to maximize the reduction in contrail warming while minimizing the number of required flight deviations.
  - Avoiding any persistent contrail requires accurate prediction of contrail formation and ice supersaturation, but strategies involving the prioritization of specific contrails by warming rely on the availability of accurate contrail models. At present, the model most widely used for this purpose is CoCiP (Schumann, 2012). CoCiP simulates the contrail cross section as a descending Gaussian plume. It is computationally efficient, open-source, and has been used in ~50% of all relevant academic works (see Appendix E). However, noting the difficulty of directly observing aged contrails, CoCiP has mostly been

calibrated against observations of young contrails (Schumann et al., 2017). There are also few evaluations of its performance against models with more complex contrail representations, such as APCEMM (Fritz et al., 2020). The lack of model intercomparison in the literature has been previously raised as a potential cause for concern (Teoh et al., 2024).

Since models have different criteria to determine contrail lifetime, understanding how long a contrail will survive for under different meteorological conditions is non-trivial. Consider long-lived contrails, which produce lower local radiative forcing (RF) over larger areas. Using data from Fig. 4(a) from Wolf et al. (2023) it can be shown that, out of two contrails of different width but the same length, depth and total ice mass, the wider contrail has a higher energy forcing than the narrower contrail (see Appendix G). Disagreement over contrail lifetime could therefore imply disproportionate disagreement in total climate impact. Furthermore, contrail lifetime is often underestimated because thin contrails are harder to detect from satellites. It is hence possible that a significant proportion of the time-integrated total extinction (a proxy for climate impact) remains unaccounted for when relying on observations, as shown by the illustrative example in Fig. 1.

Meanwhile, long-lived contrails are challenging to simulate in gridded models. A study by Dickson et al. (2009) found that 53 % of the ISSRs they observed were between 100 and 1500 m deep, and the large eddy simulations conducted in Lewellen (2014) had widths of ~50 km in the transversal direction (defined to be along the horizontal plane perpendicular to the flight direction). Furthermore, a single flight was identified as responsible for creating a cirrus cloud with a bounding box width of 130 km (measured from Fig. 12 (c) in the study by Haywood et al. (2009)). Therefore, the largest persistent contrails can reach cross-sections of up to ~100 km² in the transversal direction, making gridded simulations of sufficient resolution computationally expensive. With the high potential for climate impact and the difficulty of direct observation, model intercomparison of the predicted lifetime and time-integrated total extinction can indicate which conditions produce consistent outcomes and which need additional research.



A study employing full-lifetime large eddy simulations (Lewellen, 2014) compared its findings with a prior similar study (Unterstrasser and Gierens, 2010a – UG10a; Unterstrasser and Gierens, 2010b – UG10b), and found that "some of the inferences given in UG10a and UG10b are not supported by the present study" and "several of the parameter dependencies discussed here were found previously in UG10a and UG10b" (Lewellen, 2014). Specifically, both studies determined that the total extinction (a proxy climate impact metric) increases with the relative humidity, temperature, and initial contrail ice number. However, they found different parameters dominating the changes in total extinction: relative humidity in Unterstrasser and Gierens (2010a and 2010b), and shear in Lewellen et al. (2014) and Lewellen (2014). Since two models of similar complexity found different dominant factors in predicting a proxy for contrail climate impact, this suggests the need for a more comprehensive assessment of the robustness of contrail modelling techniques being used to inform contrail impact mitigation.

Comparisons between other models exist but are similarly limited. APCEMM and CoCiP have been previously compared by Akhtar Martinez and Jarrett (2024) and Xu (2024), showing some differences between the behavior of the models. However, no study to date has established the extent of the agreement between CoCiP and any higher-fidelity contrail model when

considering the effect of variation in meteorological parameters or the degree of consistency in identifying which contrails will have the greatest climate impact.

This study addresses this gap through a structured comparison between CoCiP and APCEMM which establishes the degree of agreement in predictions of time-integrated total extinction when six meteorological parameters are varied in full-lifetime simulations. The potential causes of the disagreements are investigated using arguments based on the physical understanding of contrail formation, persistence, and demise.

Figure 1: (a) Plan view contour plot of 1-transmittance (ratio of the light absorbed and reflected over the light received). (b) Width average vertical optical depth against time since formation. (c) Total extinction (E) against time since formation. The x axis can be interpreted as time (bottom) or as axial distance from the aircraft (top) if a constant aircraft speed is assumed. The vertical dashed line indicates the end of the unrestrained sub-regime (see Sect. 3.1). The vertical dotted line indicates the time where the contrail crosses the observability threshold, taken to be an optical depth of 0.1 (Kärcher et al., 2009). This contrail was simulated in APCEMM using the default meteorology in this study (see Sect. 2.1). This figure is only for illustrative purposes, and does not have statistical significance.

#### 1.1 Persistent contrails and contrail models


Contrails form in the aircraft plume when the Schmidt-Appleman criterion is satisfied (Schumann, 2012). For a contrail to persist, there must be ambient supersaturation with respect to ice. There are four regimes that a persistent contrail will experience throughout its lifetime: jet, vortex, dissipation, and diffusion (Gerz et al., 1998). Contrails that persist until the

diffusion regime can spread up to ~40 km horizontally (Schumann et al., 2017) and hence have the potential to have a disproportionate climate impact. Teoh et al. (2024) shows that 10 % of flights which form persistent contrails (2 % of all flights) account for 80 % of the global annual energy forcing from contrails. This implies that the most warming contrails produce the majority of their climate impact in the diffusion regime. For this reason, this investigation only considers the models in the diffusion regime.

Figure 2: Cross-sectional distribution of the ice water content (IWC) for the CoCiP plume (a), and APCEMM plume (b) ~1h 30m after formation. Both simulations use the same meteorological conditions.

#### 1.1.1 CoCiP




90

The Contrail Cirrus Prediction model (CoCiP) consists of a wake vortex sub-model, a Lagrangian Gaussian plume model, and a radiative balance model (Schumann, 2012). The CoCiP wake vortex sub-model initializes the Lagrangian Gaussian plume model in the diffusion regime, in which the concentration of ice water is assumed to follow a Gaussian distribution in space (see Fig. 2(a)). The angle and standard deviations of the resulting ellipse are modified to simulate the effect of wind shear and diffusion, while the centroid of the ellipse descends to simulate the effect of ice crystal settling. The ambient conditions with which the plume interacts are treated as being uniform in space at each centroid position. For this reason, CoCiP can be referred to as a 0D model. For a given value of vertical wind shear, the area of the simulated cross section will increase until the contrail disappears. Contrail demise in CoCiP occurs when the centroid ambient relative humidity with respect to ice falls below one or, in rare cases, when the contrail experiences excessive heating as it falls (see Appendix F). All ice crystal microphysics are represented through changes in two parameters: the total number of ice crystals (in count / m depth) and the total ice mass (in kg / m depth). The ice crystal size is a single value calculated from these quantities (Schumann, 2012), and is treated as being uniform across the contrail (i.e. the size distribution is monodisperse at any given time). The monodisperse assumption is used when calculating the ice particle size, fall speed, and optical depth of the Gaussian plume. However, ice crystal loss due to aggregation is also modelled in CoCiP (see Eq. 52 in Schumann, 2012), for

which the width of the size spectrum is assumed to be of order r (Schumann, 2012).). Although the crystal loss parameterizations implicitly assume a size distribution, all crystals are treated identically with a single radius value and no size distribution is diagnosed. As such, we still refer to CoCiP as a monodisperse model. The CoCiP version used is from the pycontrails python library, an open-source project developed and maintained by members of Breakthrough Energy and Imperial College London (details in Appendix A).

#### 115 **1.1.2 APCEMM**

120

The Aircraft Plume Chemistry, Emissions, and Microphysics Model (APCEMM) is "a Lagrangian model that explicitly models the chemical and microphysical evolution of an aircraft plume" (Fritz et al., 2020). APCEMM begins the contrail simulation with a box model for its vortex regime and then uses a 2D rectilinear grid to represent the contrail cross-section in the diffusion regime. This allows the APCEMM plume to take any shape (Fig. 2(b)). The transport and microphysical processes are computed at each grid cell. However, this grid is dynamic, meaning that the number of grid cells and the grid size change as the contrail expands. APCEMM uses a 38-bin sectional representation to approximate the ice particle size distribution in each grid cell. These bins are fixed in radius space, but the modal radius of each bin is allowed to increase within the bin bounds to accommodate the increase in ice crystal sizes with time.

APCEMM also represents ice particle loss mechanisms at a grid-cell level rather than at a contrail level (Fritz et al., 2020).

Mesoscale turbulent temperature fluctuations were not present in the original version of APCEMM but are now simulated using random temperature fluctuations. At each timestep, APCEMM disturbs the temperature in each grid cell by a random value distributed within the range [-1, 1] K. This emulates the method used in Lewellen et al. (2014) to maintain ambient turbulence. The perturbation amplitude is left at 1 K throughout this investigation, and an initial seed of 0 has been chosen for all simulations to ensure reproducibility of the results.

#### 130 2 Experimental design

Figure 3: Flow chart showing the experimental simulation process for both CoCiP and APCEMM. Green ellipses indicate inputs, blue rectangles indicate plume models, orange rhombi indicate outputs, and purple rectangles indicate data processing units.

Figure 3 provides a flow chart overviewing the experimental process in this investigation. First, the model parameters are harmonized prior to the simulations (see Appendix C). Meteorological inputs for each model are then produced to represent each simulation scenario, each of which described by six independent meteorological parameters (Section 2.1). We first compare results for a single representative scenario (Section 3), focusing on integral quantities such as total ice mass, average vertical optical depth, and time-integrated total extinction. We then quantify the sensitivity of these integral quantities to different values of six different meteorological variables (Section 4).

#### 140 2.1 Description of the background meteorology

145

All simulations use an idealized vertical weather profile consisting of a stable trapezoidal moist layer (Fig. 4), fully described by six independent meteorological variables. Ranges and default values for the six parameters are given in Table 1.

The values for the relative humidity with respect to ice saturation (RHi) in the moist layer are chosen to be above 100 % to ensure persistent contrail formation, with the highest layer RHi set to 140 % to remain beneath the homogeneous crystallization threshold of ~145 % RHi (Unterstrasser and Gierens, 2010a). The simulated range of supersaturated layer depths is chosen based on radiosonde data from the UK which showed that 53% of ice supersaturated regions are between

100 and 1500 m deep (Dickson et al., 2009). The transition between the super- and subsaturated regions is modelled with a constant gradient in RHi. The range of temperatures at cruise altitude is chosen based on estimates of typical values for northern hemisphere ice supersaturated regions (Spichtinger et al., 2003). We assume a uniform lapse rate of -6.5 K / km based on the International Standard Atmosphere. Finally, wind shear values are chosen to be consistent with the bounding scenarios of Unterstrasser and Görsch (2014). Each weather parameter is varied individually, with simulations performed at three parameter values. Contrails are not allowed to persist beyond 24 h, and no effects in the flight direction are considered.

150

155

Figure 4: An example of a parametrized weather profile used in this investigation.

Table 1: Values of the weather parameters used in this investigation. Only one parameter is varied at a time. Parameters not being varied take their baseline value, marked by an asterisk. The transition gradient at the default meteorology is infinite, making the moist layer rectangular in RHi.

| Parameter            | Units             | Low Value   | Medium Value | High Value |
|----------------------|-------------------|-------------|--------------|------------|
| Background RHi       | %                 | 20          | 40*          | 60         |
| Layer RHi            | %                 | 110         | 125*         | 140        |
| Moist Layer Depth    | m                 | 500         | $1000^{*}$   | 1500       |
| Transition Gradient* | % m <sup>-1</sup> | 0.2         | 0.4          | 0.6        |
| Temperature          | K                 | 208.15      | 215.65       | 223.15*    |
| Wind Shear           | $s^{-1}$          | $0.002^{*}$ | 0.004        | 0.006      |

# 160 2.2 Output processing and metrics

We use time-integrated total extinction  $\hat{E}$  as a proxy for climate impact. As defined in a recent manuscript by Lottermoser and Unterstrasser (2025), it is calculated as the integral over time and contrail width of the vertical optical depth in meter-hours (m h), such that

$$E(t) = \int (1 - e^{-\tau}) dx \approx \int \tau_{\nu}(x, t) dx \,, \tag{1}$$

$$\widehat{\mathbf{E}} = \int \mathbf{E}(\mathbf{t})d\mathbf{t},\tag{2}$$

where  $\tau_y$  is the vertical optical depth at a particular width coordinate x of the contrail cross-section, and E is the total extinction, as defined in Unterstrasser and Gierens (2010a) and Unterstrasser and Gierens (2010b). Given that we do not consider effects along the flight direction, the time-integrated total extinction accounts for persistence, lateral spread, and optical properties, making it implicitly related to climate impact. The time-integrated total extinction is also directly computable from both models without requiring any assumptions regarding optical properties, local cloud cover, time of day, or any of the other parameters which would need to be defined for a radiative transfer calculation.

To evaluate the effects of each weather parameter on the time-integrated total extinction, we define the sensitivity  $\Phi$  as

$$\hat{\Gamma} = \frac{\hat{E}_3 - \hat{E}_1}{\hat{E}_{mid}},\tag{3}$$

$$\Phi = \frac{\hat{\Gamma}}{\lambda_2 - \lambda_1},\tag{4}$$

where  $\hat{\Gamma}$  is the change in time-integrated total extinction throughout the sweep of a single parameter normalized by the time-integrated total extinction ( $\hat{E}_{mid}$ ) computed at the central parameter value, and  $\lambda$  is the value of the varied weather parameter at each simulation.

Due to the order of magnitude changes in total ice number throughout the contrail lifetime, we define the ice crystal loss rate as  $-\frac{d \log_{10} N}{dt}$  in decades per hour.

#### 180 3 Comparison of CoCiP and APCEMM for the baseline case

We first simulate a baseline meteorological scenario in both CoCiP and APCEMM. Results are provided in Section 3.1, followed by discussion and analysis in Section 3.2.

#### 3.1 Results






Figure 5(a) shows the simulated contrail evolution in both CoCiP and APCEMM. The time-integrated total extinction of this contrail in APCEMM (15500 m h) is four times that simulated by CoCiP (4000 m h). Additional metrics are provided in Appendix D.

The two models show qualitatively different behavior over the course of the lifetime of the contrail. The CoCiP simulation exhibits a single sub-regime which we refer to as the "unrestrained" sub-regime. This phase is defined by continuous downwards acceleration as the settling crystals encounter unperturbed air, grow (albeit sharing this growth across all crystals in the contrail), accelerate, and reach new unperturbed air, with an average fall rate of 130 m h<sup>-1</sup> until the contrail abruptly disappears upon reaching the subsaturated air after 4 hours. The total number of ice crystals in the contrail per unit depth N (Fig 5c) decreases steadily at a rate of 0.21 decades per hour throughout the entire simulation.

The APCEMM simulation shows similar behavior initially (~0–1 h), but with a small crystal loss rate of 0.05 decades per hour. It also ends sooner, as the lowermost crystals take up water and accelerate faster than the contrail-wide acceleration in CoCiP. However, rather than sublimating entirely at the end of this period, the descent rate of the remaining contrail slows. This defines the beginning of a second, "restrained" sub-regime (~1–10 h) characterized by loss of ice crystals at a rate of 0.19 decades per hour, like that shown in CoCiP. After 10 hours the contrail enters a third, "fading" sub-regime (~10–15 h) characterized by the loss rate increasing to 0.59 decades per hour, which ends with the complete demise of the contrail. From the unrestrained to the restrained sub-regimes, the average fall speed of the APCEMM center of mass fall decreases from 147 m h<sup>-1</sup> to 23 m h<sup>-1</sup>, while the fall rate during the fading sub-regime approaches 0 m h<sup>-1</sup>.

These differences are also reflected in the evolution of total contrail ice mass per unit length (Fig. 5(d)). Ice mass in CoCiP grows exponentially, whereas ice mass increase in APCEMM is closer to linear. In both models the total ice mass approaches a maximum at the end of the unrestrained sub-regime.

Through integration of Fig. 5(b), the unrestrained sub-regime and the sub-regimes after this contribute 9 % and 91 % respectively to the APCEMM time-integrated total extinction, compared to 100 % and 0 % for CoCiP. In absolute terms, the unrestrained sub-regime contributes ~4000 m h and ~1400 m h to the CoCiP and APCEMM time-integrated total extinction respectively.

Figure 5: Evolution of contrail properties at the baseline meteorology. (a) Center of mass altitude against time. The dotted red horizontal line indicates the lower limit of the moist layer. (b) Total extinction (E) against time. (c) Total number of ice crystals (N) against time. The exponential decay fits for CoCiP and APCEMM are given by the dashed and dotted straight lines respectively. (d) Total ice mass (I) against time. The vertical lines mark the APCEMM sub-regime transition times.

#### 3.2 Discussion


The differences in model behaviors observed in Fig. 5 can be explained by considering the contrail representations of each model. CoCiP defines its contrail properties at the center and uses a monodisperse ice crystal radius distribution, whereas APCEMM discretizes space into ~350 m<sup>2</sup> grid cells and uses 38 ice radius bins at each of these cells. This allows for ice crystals of different sizes to fall at different rates and separate spatially in APCEMM. Like in Lewellen (2014), the APCEMM contrail can be simplified into two components: "a core near flight level with larger number densities and a much more sparsely populated precipitation plume below with larger crystals" (Lewellen, 2014).

Mathematical definitions of the sub-regimes observed in Fig. 5 can be formulated by considering the total ice mass per unit length (*I*), with the caveat that they are only likely to be valid for contrails simulated in idealized meteorology. First, the unrestrained sub-regime is defined as the sub-regime in which the precipitation plume has not yet reached the subsaturated

layer. Throughout the unrestrained sub-regime  $\frac{dl}{dt} > 0$  and  $\frac{d^2l}{dt^2} > 0$  since water deposition outweighs sublimation. The restrained sub-regime begins when the precipitation plume first reaches the subsaturated layer and the ice crystals begin to sublimate in it, characterized by  $\frac{d^2l}{dt^2} \le 0$ . The fading sub-regime begins when the contrail core reaches the subsaturated layer. During the fading sub-regime  $\frac{dl}{dt} < 0$  and  $\frac{d^2l}{dt^2} > 0$  since the sublimation rate decreases progressively as fewer ice crystals remain.

Since CoCiP cannot represent this differential sedimentation (see Sect. 4.2.2), which is typical of cirrus clouds (Sölch and Kärcher, 2010), it cannot represent the separation of the precipitation plume from the contrail core. CoCiP simulations therefore capture only behavior in the unrestrained sub-regime according to the above mathematical definitions (albeit with some anomalies due to certain model assumptions - see Appendix F). With this, the evolution of the remaining contrail properties in each sub-regime can now be considered.





During the unrestrained sub-regime ice crystals grow rapidly in both models leading to the largest center of mass fall rate (Fig. 5(a)). In CoCiP, the entire contrail moves with the center due to its monodisperse ice radius distribution. In APCEMM, the ice radius bins allow the large particles to fall the fastest, a form of gravitational size sorting which results in the formation of a precipitation plume beneath the contrail core. An enhancement to vertical diffusivity is applied to try and compensate for the absence of this gravitational sorting in CoCiP (Schumann, 2012). Nonetheless, in CoCiP the descent rate of the plume is lower, the unrestrained sub-regime lasts longer, and the contrail acquires ice mass at a slower rate than in the unrestrained sub-regime in APCEMM.

When the large ice particles exit the supersaturated layer they sublimate, making the contrail lose ice mass. In CoCiP this results in near-instantaneous demise, as the entire contrail experiences the same conditions. In APCEMM this instead results in a decrease in the center of mass fall rate from 147 m h<sup>-1</sup> to 23 m h<sup>-1</sup> (Fig. 5(a)), marking the start of the restrained subregime. Once the contrail core reaches the subsaturated region it enters the fading sub-regime, during which only the ice particles with the lowest fall rate remain. This results in a progressive decay in the center of mass fall speed and eventual contrail demise. A video of the contrail evolution simulated in APCEMM can be found in the Video Supplement.

Although some equations in CoCiP have been calibrated (Shapiro et al., 2024) to fit observations from the Contrails Library (COLI) database (Schumann et al., 2017), these adjustments cannot represent the transition of the contrail to a new subregime due to the monodisperse Gaussian plume assumption. Observations of long-lived contrails are also rare due to their low average vertical optical depth. For the baseline case, 25 % of the time-integrated total extinction in APCEMM is produced at times when the average vertical optical depth is below 0.1, meaning that the aged APCEMM contrail would likely not be observable from satellites (Kärcher et al., 2009). Properties and behaviour in the unobservable region are therefore particularly difficult to calibrate from observational data.

We estimate that ~91 % of the time-integrated total extinction is produced beyond the unrestrained sub-regime in APCEMM, whereas this figure is 0 % for CoCiP (Fig. 5(b)) for the baseline case. Similar behavior to APCEMM has been observed in studies employing large eddy simulations (Unterstrasser and Gierens, 2010a, Lewellen et al. 2014, Lewellen, 2014). This

does not imply the correctness of the APCEMM, but it does suggest that some behavior beyond the unrestrained sub-regime should be expected for long-lived contrails.

#### 3.2.1 Implications for real contrails





The presence of the three distinct sub-regimes could have resulted from the controlled experimental setup relying on time-stable ISSRs, making it important to consider their typical lifetimes. Schumann (2012) analyzed ECMWF data for 6–9 June 2006 and found that 1 % of ISSRs had lifetimes over 24 h. Irvine et al. (2014) conducted a study tracking the advection of ISSRs in the North Atlantic using ECMWF Interim reanalysis data from three winter and summer seasons. Their results show that the mean lifetime (in a Lagrangian sense) for ISSRs is ~6 h, and that 5 % of ISSRs forming in the troposphere will have lifetimes exceeding 24 h, with their Fig. 4(a) indicating that the proportions of wintertime tropospheric ISSRs persisting for over 6, 12, 18, and 24 h are 32, 14, 6, and 2 % respectively. Overall, these studies indicate that time-stable ISSRs account for a non-negligible proportion of all ISSRs.

A recent preprint by Hofer and Gierens (2024) analyzed the ICON dataset and found that contrail lifetime is most commonly limited by sedimentation and synoptic processes such as advection of contrails out of the ISSRs, or subsidence. Furthermore, a second preprint by Hofer and Gierens (2025) found that the typical sedimentation and synoptic timescales are both approximately up to ~8 h each. This is corroborated by a study estimating the full-lifetime of contrails with statistical methods applied to satellite observations (Gierens and Vazquez-Navarro, 2018). Interpolating Fig. 8 from Gierens and Vazquez-Navarro (2018), it can be estimated that the proportion of contrails with lifetimes exceeding 8 h is ~6–7 %. This implies that the characterization of the sub-regimes observed in this study may be applicable to some long-lived persistent contrails, likely including some of the contrails that are responsible for 80 % of the climate impact (Teoh et al., 2024).

# 275 4 Effects of varying weather parameters

We now simulate 14 different meteorological scenarios, spanning variations in six different weather parameters (Table 1). We first compare the general trends in the time-integrated total extinction (Section 4.1.1) and then the sensitivity with respect to each parameter (Section 4.1.2).

#### 4.1 Results

#### 280 4.1.1 Time-integrated total extinction

Figure 6(a) compares the time-integrated total extinction from CoCiP and APCEMM when considering all contrail lifetime (orange) and when only considering the unrestrained sub-regime (purple).

The CoCiP and APCEMM simulations disagree regardless of whether the entire lifetime or the unrestrained sub-regime are considered in isolation. When only the APCEMM unrestrained sub-regime is considered, CoCiP simulations have time-integrated total extinction values 3.3 times larger than those from the corresponding sub-regime in APCEMM (given by the

reciprocal of the slope of the purple dashed line in Fig. 6(a)). The case in which all sub-regimes are considered lies above the parity line, with APCEMM simulations having a time-integrated total extinction 3.8 times that of CoCiP (given by the slope of the orange dotted line in Fig. 6(a)).

The relationship between the proportion of the time-integrated total extinction in the unrestrained sub-regime and unobservable regions is displayed in Fig. 6(b). Considering the following sums across all 14 unique simulations:

$$\sigma = \frac{\sum_{all\ cases}(\hat{E}_{model}(t=t*))}{\sum_{all\ cases}(\hat{E}_{model}(t=24\ h))},\tag{5}$$

where t\* is a chosen integration threshold, we find that 92 % of APCEMM time-integrated total extinction is produced after the unrestrained sub-regime, and 38 % is produced when the contrail is unobservable. In contrast, across all simulations CoCiP produces none of its time-integrated total extinction beyond the unrestrained sub-regime, and 17 % beyond the observability threshold.

#### 4.1.2 Sensitivity to weather parameters





Despite the differences in the baseline case, the models mostly agree on the sign of the sensitivity of the time-integrated total extinction with regards to the six meteorological parameters (Table 2, Fig. 7). The exception is the sensitivity to moist layer RHi, where CoCiP finds a 1.2 % reduction in time-integrated total extinction per percentage point increase in RHi compared to a 3.4 % increase in APCEMM. Although there is some anomalous behavior in CoCiP for the 110 % RHi case (see Figure 8 and Appendix F), the disagreement in sign remains even when this case is excluded.

Otherwise, both models show a decrease in  $\Gamma$  with increasing temperature and an increase in  $\Gamma$  with increases in all other parameters. The sign of the sensitivity is consistent whether considering the full APCEMM lifetime or only the unrestrained sub-regime. The largest disagreement in the value of the sensitivity between APCEMM and CoCiP for the parameters where the sign agrees is in wind shear (10 % per m/s/km for APCEMM, compared to 5.1 % in CoCiP). Figure 9 demonstrates this in terms of the effect of wind shear on altitude in each simulation, with settling velocities increasing with growing wind shear in CoCiP. In APCEMM, the settling velocity is unaffected during the unrestrained sub-regime but is similarly increased by increased wind shear after the unrestrained sub-regime . Furthermore, at the end of the APCEMM unrestrained sub-regime, the higher shear leads to an increase of the contrail width by 143 % and an increase of the ice mass by 58 %, leading to a 36 % increase of the time-integrated total extinction between the 0.002 s<sup>-1</sup> and the 0.006 s<sup>-1</sup> shear cases (see Appendix D).

Figure 6: (a) Parity plot for the time-integrated total extinction ( $\hat{E}$ ). The dotted line is the line of best fit for all sub-regimes, and the dashed line is the line of best fit for the unrestrained sub-regime only. (b) Parity plot for the unobservable proportion of time-integrated total extinction. The solid black line indicates the line of equality. Each entry in the parity plot corresponds to one simulation. In (a), the crosses indicate the simulations where the whole lifetime has been considered, whereas the dots indicate the simulations where only the unrestrained sub-regime has been considered.

Figure 7: Bar chart showing the relative percentage change in time-integrated total extinction across the contrail-producing simulations ( $\hat{\Gamma}$ ) for each meteorological variable. A positive value indicates that an increase in a particular variable leads to an increase in the contrail time-integrated total extinction.

Table 2: Sensitivity ( $\Phi$ ) of each weather parameter. A positive value of  $\Phi$  indicates that increasing the parameter yields an increase in the time-integrated total extinction ( $\hat{E}$ ). The APCEMM sensitivities to the temperature are underestimates because the contrail persists beyond the maximum simulation time of 24 h.

|           |              | APCEMM unrestrained |                   |         |
|-----------|--------------|---------------------|-------------------|---------|
| Parameter | $\Phi$ units | $APCEMM \Phi$       | sub-regime $\Phi$ | CoCiP Φ |


| Background RHi      | $\%_{\text{change}}(\hat{E}) \cdot (\text{RHi \%})^{-1}$                       | 0.14  | 0.0024 | 0.070 |
|---------------------|--------------------------------------------------------------------------------|-------|--------|-------|
| Layer RHi           | $\%_{\text{change}}(\hat{E}) \cdot (\text{RHi \%})^{-1}$                       | 3.4   | 0.63   | -1.2  |
| Moist Layer Depth   | $\%_{\text{change}}(\widehat{E}) \cdot \text{km}^{-1}$                         | 77    | 99     | 68    |
| Transition Gradient | $\%_{\text{change}}(\widehat{E}) \cdot (\text{RHi \% km}^{-1})^{-1}$           | 0.080 | 0.28   | 0.10  |
| Temperature*        | $\%_{\text{change}}(\widehat{E}) \cdot \mathrm{K}^{\text{-}1}$                 | -4.0  | -2.0   | -3.7  |
| Wind Shear          | $\%_{\text{change}}(\widehat{E}) \cdot (\text{m s}^{-1} \text{ km}^{-1})^{-1}$ | 10    | 9.5    | 5.1   |

Figure 8: Total extinction (E) against time for the contrails produced at varying layer RHis. The CoCiP and APCEMM simulations are represented by solid and dotted lines respectively.


Figure 9: Center of mass altitude against time for the contrails produced at varying wind shear. The CoCiP and APCEMM simulations are represented by solid and dotted lines respectively.

#### 4.1.3 Effect of the contrail lifetime horizon on the model disagreement


#### 335 Figure 10: Normalized global model difference $\hat{\delta}(t)$ in the time-integrated total extinction as a function of time since formation.

To understand the sensitivity of our findings to the contrail lifetime, we define the global model difference ( $\delta$ ) as the sum across all simulations of the APCEMM integrated total extinction minus the CoCiP integrated total extinction (at each timestep):

$$\delta(t) = \sum_{all\ cases} (\hat{E}_{APCEMM}(t) - \hat{E}_{CoCiP}(t)), \tag{6}$$

$$\hat{\delta}(t) = \frac{\delta(t)}{\delta(t = 24 \text{ h})},\tag{7}$$

where  $\hat{\delta}(t)$  is the normalized global model difference. The variable t in Eqs. 5 and 6, is the upper limit of integration in Eq. 2.

Figure 10 shows how  $\hat{\delta}(t)$  varies as a function of time. We hence find that 90 % of the global model difference is produced within 12 hours from formation. For more evidence-based contrail lifetime estimates, we take 4 h and 8 h from a recent preprint by Hofer and Gierens (2025). The proportion of the total model difference reached by 4 h and 8 h are 27 % and 72 % respectively. These results indicate a large sensitivity in our findings to the lifetime of typical contrails. However, they also indicate that our findings are particularly relevant to those 6–7 % of contrails that persist beyond 8 h (Gierens and Vazquez-Navarro, 2018). Such contrails are also likely to be the greatest contributors to aviation warming on an individual basis and are hence important for contrail avoidance.

#### **4.2 Discussion**





# 4.2.1 Extent of agreement in predicted time-integrated total extinction

As in the baseline case (Section 3.1), APCEMM consistently predicts a shorter unrestrained sub-regime followed by longer sub-regimes beyond this when compared to CoCiP. This explains why the CoCiP contrails last ~3 times longer than the corresponding APCEMM contrails in their unrestrained sub-regimes, with time-integrated total extinctions on average 3.3 times greater (Fig. 6(a)). This ratio reverses when the full contrail lifetime is considered, however, and Fig. 6(b) shows that a higher proportion of time-integrated total extinction also occurs in APCEMM than in CoCiP after the contrail becomes "unobservable" (38 % and 17 % respectively). If post-unrestrained and post-observable behaviors are as important as suggested by APCEMM, approximations may be needed to extend 0D modelling techniques to the restrained and fading sub-regimes.

Since almost all CoCiP contrails evaporate when the plume center reaches the subsaturated layer, the CoCiP contrails with large initial ice mass densities are much less likely to reach the observability threshold than the equivalent APCEMM contrails because they experience total demise more prematurely than the cases with low initial ice mass densities. This suggests that CoCiP might underestimate the climate impact of the contrails we already deem to be the most impactful. Nevertheless, further research is necessary to characterize the accuracy of the climate impact predictions from both CoCiP and APCEMM.

#### 4.2.2 Extent of agreement in sensitivity

Increasing the layer RHi causes the time-integrated total extinction to decrease in CoCiP and increase in APCEMM. Fig. 1 in Lewellen (2014) shows that increasing the layer RHi increases the total ice crystal count, ice mass, and ice particle surface area throughout the contrail lifetime. Similarly, Fig. 4 and Fig. 6 in Unterstrasser and Gierens (2010a) also confirm that increasing the layer RHi increases the ice mass and the total ice crystal count respectively. Since the optical depth can be thought to increase with the contrail ice mass and ice surface area, it can be deduced that increasing the layer RHi would result in an increase in time-integrated total extinction, and hence a positive sensitivity to the layer RHi. The CoCiP sensitivity to layer RHi is not consistent with APCEMM, Lewellen (2014), and Unterstrasser and Gierens (2010a). This has implications for the implementation of robust contrail avoidance strategies, which require avoiding contrails with predicted characteristics agreed on by several models. Explaining disagreement in the sensitivity is hence necessary to be able to bridge the behaviors of the models.

The opposite sensitivity to the layer RHi occurs because most of the time-integrated total extinction stems from the unrestrained sub-regime for CoCiP and the post-unrestrained sub-regimes for APCEMM (see Sect. 3.2). For CoCiP, Fig. 8 shows that E grows at higher rates as the layer RHi increases. This happens due to the associated increase in contrail ice mass. However, this effect competes with the decreasing lifetime caused by the increased settling speed of the larger ice particles in the higher layer RHi cases. Overall, the lifetime shortening effect outweighs the growth rate effect in  $\hat{E}$ , leading

to a negative sensitivity to the layer RHi in CoCiP (Table 2). For APCEMM, Fig. 8 shows that E reaches larger values for higher layer RHis throughout the entire lifetime. Like in CoCiP, the unrestrained sub-regime in APCEMM terminates sooner with increasing layer RHi due to the increased rate of mass accumulation of the lowermost settling ice particles. However, the overall APCEMM lifetime increases. This makes the sensitivity for the APCEMM unrestrained sub-regime (0.63 units, Table 2)  $\sim$ 6 times lower than the sensitivity of the entire APCEMM contrail (3.4 units, Table 2). The positive sensitivities confirm that the dominating effect for both the entire APCEMM lifetime and for the APCEMM unrestrained sub-regime is the increase in E with increasing layer RHi.







increases to the settling rate during the unrestrained.

For the remaining variables, CoCiP and APCEMM both display similar sensitivity magnitudes, except for wind shear (Table 2, Fig. 7). Increasing the wind shear increases the contrail area horizontally and allows some crystals to settle into unperturbed air, hence leading to an increase in the amount of water uptake from the ambient air. The increased ice mass results in an increase in particle size, optical depth, and settling speed, although in theory this should not affect the lowermost particles which always fall through unperturbed air.

The effect of the increased settling rate due to the increased particle size competes with the greater vertical optical depth and

increased area. Other studies find qualitatively similar results to APCEMM: increasing the shear leads to higher ice masses earlier (Lewellen, 2014 and Unterstrasser and Gierens, 2010a) and lower lifetimes (Lewellen, 2014). This leads to a limited increase in  $\hat{E}$  with a moderate sensitivity to wind shear (5.1 units for CoCiP, 10 units for APCEMM, and 9.5 units for the APCEMM unrestrained sub-regime, Table 2). APCEMM is approximately twice as sensitive to shear as CoCiP because, unlike APCEMM, CoCiP shows its increased settling rate throughout its lifetime due to the 0D nature of the model (Fig. 9). The APCEMM precipitation plume falls quickly and is not exposed to much more unperturbed ambient air by the shear. This limits the water intake of the precipitation plume, limiting the increase in size of its ice particles. The lifetime of the APCEMM unrestrained sub-regime hence appears unaffected (Fig. 9). Despite this, the shear in APCEMM does increase the contrail width at the end of the unrestrained sub-regime by 143 % and the unrestrained sub-regime ice mass by 58 %, leading to a 36 % increase in time-integrated total extinction for the APCEMM unrestrained sub-regime between the 0.002 s<sup>-1</sup> and the 0.006 s<sup>-1</sup> shear cases (see Appendix D). This shows that the slower moving core has time to take up more water than the

It is also helpful to consider the effect that wind shear has on a contrail after the time when the unrestrained sub-regime ends. For a real contrail in a constant shear environment, sublimation of the ice reaching the subsaturated layer causes the contrail shape to become truncated (see Fig. 2(b)). As the uppermost contrail crystals continue to settle and reduce the vertical extent of the contrail, this leads to a reduction in the contrail widening rate. In CoCiP the contrail instead continues to steadily gain ice through deposition as long as the centroid lies above the subsaturation point, and its cross-section continues to widen. Due to its spatially distributed ice particle size spectrum, the APCEMM contrail can show behavior closer to that expected from a real contrail.

precipitation plume, leading to the observation of an increased settling rate in the restrained sub-regime with negligible

#### 415 **4.2.3** Implications for contrail avoidance




Contrail avoidance strategies that do not attempt to prioritize avoiding the most warming persistent contrails can be performed without a contrail model, needing only an estimate of whether the Schmidt-Appleman criterion and ice supersaturation have been met simultaneously. However, any further prioritization will necessarily be based on understanding the relationship between local meteorology, aircraft parameters, and the eventual contrail lifetime. Compared to simulations with APCEMM, contrails simulated in CoCiP have 3.8 times lower time-integrated total extinction, the opposite sensitivity to changes in the local relative humidity, and approximately half the sensitivity to local wind shear. In the context of robust contrail avoidance strategies, the disagreement in baseline optical depth means that, due to the lack of behavior beyond the unrestrained sub-regime, CoCiP predictions may underestimate the role of long-lived contrails including the potential for (typically cooling) daytime contrails to persist into nighttime and become warming. Meanwhile, the different sensitivities mean that the models will likely disagree regarding which contrails should be prioritized for avoidance.

The two models do agree on the sign and order-of-magnitude of the relative sensitivity to other factors such as temperature and supersaturated layer depth. However, the results shown here suggest that efforts to prioritize specific contrails based on model-simulated radiative forcings may be premature.

#### 430 5 Limitations and further work

We only consider a limited set of parameters in this work. We do not consider sensitivity to turbulence and vertical winds, which are parameters known to strongly affect contrail evolution. This means that the findings from this study cannot be generalized for all long-lived contrails. In addition, no evaluations have been performed with changes to more than one variable at a time. Furthermore, the inclusion of non-meteorological variables such as soot emissions, aircraft wingspan, and total mass would have provided further insights into the different processes captured by each model. Although sensitivity has been considered, the simulation conditions have been controlled and hence uncertainty has been neglected. Studies that extend the comparison to include model and weather uncertainty considerations are hence recommended. Finally, to determine the full extent of the climate implications of the comparison, we encourage future studies to include radiative transfer calculations on a set of contrail simulations around the globe.

#### 440 6 Conclusions

The CoCiP and APCEMM contrail models fundamentally differ in terms of how they represent a contrail. CoCiP represents the contrail as a descending Gaussian plume, efficiently approximating the early behavior. However, APCEMM predicts two additional sub-regimes after the initial unrestrained sub-regime, and our results suggest that these later sub-regimes provide 92% of the overall time-integrated total extinction. This discrepancy in contrail representation means that the two models

- predict different magnitudes and, for the local relative humidity, signs of the relationships between the time-integrated total extinction of a contrail and local meteorological parameters.
  - This work is highly idealized, considering only stable ice supersaturated regions which can support very long contrail lifetimes. However, tropospheric ice supersaturated regions are generally sufficiently large that contrail demise occurs through sedimentation, synoptic processes, or both at similar timescales (Hofer and Gierens, 2024; Hofer and Gierens,
- 2025;). Furthermore, 72 % of the model disagreement in the time-integrated total extinction can be attributed to the first 8 hours of the simulations (see Sect. 4.1.3). Since Gierens and Vazquez-Navarro (2018) found that ~6–7 % of contrails persist beyond 8 hours, this makes it likely for the conceptual findings from this study to be applicable to the real contrails with the greatest warming. Nevertheless, the observed differences between the models are not just limited to the lack of the later subregimes in CoCiP.
- This work suggests that strategies prioritizing the most warming contrails for avoidance (e.g. Teoh et al., 2020a), although relevant for thought experiments, are likely not yet realizable in practice. Although physical arguments and prior large eddy simulation results provide some evidence that APCEMM is producing a more realistic simulation than CoCiP, this does not serve to validate or discredit either model. Further research and experiments are needed to characterize full lifetime contrail behaviour. This includes the challenging period where contrails are too thin to be easily observed from satellite, which we find to be responsible for 38 % of total contrail time-integrated total extinction in APCEMM simulations. Until efficient, reliable contrail models are available and backed by such evidence, our results suggest that unbiased contrail avoidance strategies at any scale will have the greatest chance of producing a real climate benefit.

#### Appendix A: CoCiP version and modifications

CoCiP simulations on the development version of pycontrails based on v0.54.0. The code branch used for this investigation was created from the 760244d commit (dated 16<sup>th</sup> Sept 2024) in the pycontrails main branch.

The following is a list of changes made to a fork of the pycontrails repo for this investigation. Model parameters that are not mentioned have been left at a default value:

- Increased the maximum contrail age from 20 to 24 h
- Decreased the integration timestep from 30 m to 5m to match APCEMM
- Removed the shear enhancement factor
- Force total and normal shear to take either of the following values depending on the case: 2e-3, 4e-3, or 6e-3 s<sup>-1</sup>.
- Horizontal advection was disabled





- Forced the Brunt-Väisälä frequency to be 0.01 s<sup>-1</sup> instead of being calculated from the meteorology
- Set the maximum contrail depth to infinity

See the Code Availability section for access to the code.

#### **Appendix B: APCEMM version and modifications**

APCEMM simulations on the development version based on v1.2.0. The branch of code used for this investigation was created from the c19b7f3 commit (dated 6<sup>th</sup> November 2024) in the APCEMM main branch. Modifications to APCEMM were made to enable the user selection of the random number generation seed.

See the Code Availability section for access to the code.

#### Appendix C: Aircraft and flight parameters

The aircraft used for this investigation was the Boeing 737-800. Table C1 shows the aircraft and flight parameters used in each model. Where no deterministic 1:1 relationship exists between equivalent parameters in each model, the closest default value was used.

Table C1: Aircraft and flight parameters used in this investigation. The values are estimates for a Boeing 737-800.

| Flight Parameter Name | Value | Units             | In pycontrails? | In APCEMM? | Notes                |
|-----------------------|-------|-------------------|-----------------|------------|----------------------|
| Cruise altitude       | 10000 | m                 | ✓               | ✓          | 264.36 hPa in APCEMM |
| Cruise speed          | 240   | m s <sup>-1</sup> | ✓               | ✓          |                      |
| Mach number           | 0.80  |                   | ✓               |            | Assumed constant     |
| Brunt-Väisälä freq.   | 0.01  | Hz                | ✓               | ✓          |                      |

| Soot EI               | 0.08                 | $g kg^{-1}$        |   | $\checkmark$ | Equivalent to nvPM EI              |
|-----------------------|----------------------|--------------------|---|--------------|------------------------------------|
| Soot radius           | 20.10-9              | m                  |   | ✓            |                                    |
| nvPM EI               | $1.19 \cdot 10^{15}$ | # kg <sup>-1</sup> | ✓ |              |                                    |
| Total fuel flow       | 0.70                 | kg s <sup>-1</sup> | ✓ | $\checkmark$ |                                    |
| Number of engines     | 2                    |                    | ✓ | $\checkmark$ |                                    |
| Wingspan              | 34.32                | m                  | ✓ | $\checkmark$ |                                    |
| Wing area             | 124.6                | $m^2$              | ✓ |              |                                    |
| Exit bypass area      | 0.9772               | $m^2$              |   | $\checkmark$ |                                    |
| Engine efficiency     | 0.295                |                    | ✓ |              | No 1:1 equivalence with exit temp. |
| Core exit temperature | 553.65               | K                  |   | $\checkmark$ | No 1:1 equivalence with efficiency |
| Aircraft mass         | 60000                | kg                 | ✓ | $\checkmark$ |                                    |
| SO <sub>2</sub> EI    | 1.20                 | $g kg^{-1}$        | ✓ | $\checkmark$ |                                    |
| Latitude              | 52.1983              | 0                  | ✓ | $\checkmark$ |                                    |
| Longitude             | 0.1202               | 0                  | ✓ | $\checkmark$ |                                    |

# Appendix D: Tabulated time-integrated total extinction results


Table D1: Time-integrated total extinction for each model simulation that varies the background RHi.

|                   | APCEMM unrestrained        |                            |              |  |  |
|-------------------|----------------------------|----------------------------|--------------|--|--|
| Background RHi, % | APCEMM $\widehat{E}$ , m h | sub-regime $\hat{E}$ , m h | CoCiP Ê, m h |  |  |
| 20                | 15224                      | 1434                       | 4032         |  |  |
| 40                | 15497                      | 1434                       | 4044         |  |  |
| 60                | 16112                      | 1435                       | 4146         |  |  |

Table D2: Time-integrated total extinction for each model simulation that varies the layer RHi.

|              |               | APCEMM unrestrained        |              |
|--------------|---------------|----------------------------|--------------|
| Layer RHi, % | APCEMM Ê, m h | sub-regime $\hat{E}$ , m h | CoCiP Ê, m h |
| 110          | 6890          | 1330                       | 4854         |
| 125          | 15497         | 1434                       | 4044         |
| 140          | 22794         | 1602                       | 3382         |

Table D3: Time-integrated total extinction for each model simulation that varies the moist layer depth.

|                      |               | APCEMM unrestrained |              |
|----------------------|---------------|---------------------|--------------|
| Moist Layer Depth, m | APCEMM Ê, m h | sub-regime Ê, m h   | CoCiP Ê, m h |
| 500                  | 9302          | 372                 | 2552         |
| 1000                 | 15497         | 1434                | 4044         |
| 1500                 | 21290         | 1785                | 5293         |

# 495 Table D4: Time-integrated total extinction for each model simulation that varies the transition gradient. The mid-range parameter value is 0.6 % m<sup>-1</sup>.

|                                        | APCEMM unrestrained   |                            |              |  |  |
|----------------------------------------|-----------------------|----------------------------|--------------|--|--|
| Transition Gradient, % m <sup>-1</sup> | $APCEMM \hat{E}, m h$ | sub-regime $\hat{E}$ , m h | CoCiP Ê, m h |  |  |
| 0.2                                    | 9890                  | 173                        | 2263         |  |  |
| 0.4                                    | 13370                 | 852                        | 3455         |  |  |
| 0.6                                    | 14144                 | 1126                       | 3707         |  |  |
| $\infty$                               | 15497                 | 1434                       | 4044         |  |  |

Table D5: Time-integrated total extinction for each model simulation that varies the temperature.

|                | APCEMM unrestrained   |                            |              |  |  |
|----------------|-----------------------|----------------------------|--------------|--|--|
| Temperature, K | $APCEMM \hat{E}, m h$ | sub-regime $\hat{E}$ , m h | CoCiP Ê, m h |  |  |
| 208.15         | 27494                 | 1892                       | 7449         |  |  |
| 215.65         | 19748                 | 1524                       | 6114         |  |  |
| 223.15         | 15497                 | 1434                       | 4044         |  |  |

# 500 Table D6: Time-integrated total extinction for each model simulation that varies the wind shear.

|                             | APCEMM unrestrained   |                                |              |  |  |
|-----------------------------|-----------------------|--------------------------------|--------------|--|--|
| Wind Shear, s <sup>-1</sup> | $APCEMM \hat{E}, m h$ | sub-regime $\widehat{E}$ , m h | CoCiP Ê, m h |  |  |
| 0.002                       | 15497                 | 1434                           | 4044         |  |  |
| 0.004                       | 19489                 | 1504                           | 4549         |  |  |
| 0.006                       | 23625                 | 2003                           | 4971         |  |  |

#### Appendix E: Prominence of CoCiP and APCEMM in academic works

Two metrics were used to determine the prominence of CoCiP and APCEMM: the citation number for the relevant model articles and the number of academic works either model has been mentioned in. Both metrics were extracted from Google Scholar.

Schumann (2012), the article which introduced CoCiP, has been cited 141 times, whereas Fritz et al (2020), the article which introduced APCEMM, has been cited 27 times as of the 25<sup>th</sup> of September 2024. CoCiP has approximately 5 times the citation count that APCEMM has.

The number of academic works that mention specific models is the number of results in a Google Scholar search with specific search terms, given below:

- Contrail models: "contrail cirrus prediction model" OR APCEMM OR "contrail model" OR "contrails model" OR "contrails model" OR "contrails models" OR "contrails model"-myuouue
  - CoCiP: "contrail cirrus prediction model"
  - APCEMM: APCEMM -myuouue


Note that the term *-myuouue* was used to remove a specific article that contained APCEMM but was unrelated to contrail modelling.

As of the 25<sup>th</sup> of September 2024, the number of mentions was as follows: 348 for contrail models, 185 for CoCiP, and 21 for APCEMM. CoCiP and APCEMM are hence mentioned in ~50% and ~5% of articles mentioning contrail models. Please note that these are only rough estimates.

#### Appendix F: Explanation of the anomalous CoCiP result at 110 % layer RHi

In Fig. 8, the CoCiP simulation at 110 % layer RHi shows a decrease in the total extinction that does not correspond to behavior beyond the unrestrained sub-regime. CoCiP has been formulated such that all the water content in the contrail (both vapor and ice) moves together with the contrail center. This water is isolated from the ambient humidity, except for the water added to the core to emulate mixing. If the ambient conditions are at or above ice saturation, the water mass in the CoCiP contrail is distributed between water vapor and ice to ensure that the air in the contrail is ice saturated. As the contrail falls, 525 the ambient temperature increases due to the atmospheric lapse rate. The higher temperature leads to an increase in the amount of water required to maintain saturation with respect to ice. Initially, mixing with ambient air supplies more water than that required to maintain saturation, so the excess water is used to grow the ice crystals. Since the saturation humidity grows in a superlinear manner with temperature, as the contrail falls progressively larger amounts of water from mixing with the ambient air are required to maintain ice supersaturation. After 5 h the increase in water required to maintain saturation is 530 no longer met by mixing. This causes the ice crystals to evaporate to maintain saturation with respect to ice, as dictated by the CoCiP formulation. This leads to a decrease in both the ice mass and total extinction and, eventually, to the contrail demise within the moist layer. This behavior is not accurate because the ice crystals fall independently of the water vapor in reality. Note that the 110 % layer RHi case has not been excluded from the results of this study because the effect is present in all the CoCiP simulations, despite it only being clearly visible in the 110 % layer RHi case (see Appendix F).

#### 535 Appendix G: The importance of contrail width to energy forcing

Consider two contrail segments under the same ambient conditions with the same ice mass per unit contrail length of 20 kg m-1, with both contrails having the same depth of 500 m. Assuming that contrail A is 1 km wide, and that contrail B is 2 km wide, the ice water content (IWC) of contrail A (0.04 g m<sup>-3</sup>) is two times the IWC of contrail B (0.02 g m<sup>-3</sup>). Fig. 4b) of Wolf et al. (2023) shows that contrail A will have an instantaneous longwave radiative forcing (LW RF) of ~125 W m-2, while contrail B will have an instantaneous LW of ~115 W m<sup>-2</sup>. Accounting for the contrail width, contrail A will have a LW RF of ~125 W m<sup>-1</sup>, whereas contrail B will have a LW RF of ~230 W m<sup>-1</sup>. Contrail A, the one with higher optical depth (due to its higher IWC), will have a lower energy forcing than the more dilute but wider contrail B. This implies that, for the same total ice mass, contrails that have a large horizontal span but are optically thin may have a greater climate impact than thicker, narrower contrails.

### 545 Appendix H: Ice particle losses in the CoCiP plume





Following the initial loss of ice crystals during vortex sinking, CoCiP assumes continuous loss of ice crystals through three mechanisms – parameterizing some of the effects of a non-monodisperse size distribution to estimate the effect on the two stored values (total ice mass and total crystal number):

• Losses due to internal plume turbulence (denoted as "turb" in Equation 49 from Schumann (2012)):

"
$$(dN/dt)_{\text{turb}} = -E_T \left( \frac{D_H}{\max(B,D)^2} + \frac{D_V}{D_{\text{eff}}^2} \right) N$$
"

- Losses due to sedimentation-induced aggregation (denoted as "agg" in Equation 52 from Schumann (2012)):  $(dN/dt)_{agg} = -E_A 8\pi r^2 V_T N^2 / A"$
- Losses by turbulent humidity fluctuations, mesoscale turbulence, and gravity waves (denoted as "meso" in Equation 55 from Schumann (2012))

"
$$(dN/dt)_{\rm meso} = -E_{\rm meso}N_{BV}w'_{\rm meso}(dT/dz)/\Delta T_c$$
"

The only other mechanism for loss of crystals is total evaporation of the contrail. Once subsaturated air begins to mix with the contrail, all crystals will give up ice to maintain 100% saturation – meaning that the ice mass (and therefore the effective radius) decreases uniformly. Once one of the end-of-life conditions is reached, all crystals are eliminated instantaneously. This can take a few time steps but is typically very rapid as CoCiP does not model the horizontal distribution of water vapor or crystals within the contrail air mass.

The equations for these losses depend on several time-varying contrail and ambient properties. Hence, even though CoCiP is a monodisperse model, the ice loss rate is not constant when the plume is in supersaturated air.

#### Code availability

The code necessary to produce the simulations and results from this investigation is available at <a href="https://doi.org/10.5281/zenodo.14708885">https://doi.org/10.5281/zenodo.14708885</a>.

The modified pycontrails code is available at https://doi.org/10.5281/zenodo.14639631.

The modified APCEMM code is available at https://doi.org/10.5281/zenodo.14640899.

#### Video supplement

A set of videos showing the evolution of the APCEMM simulated contrail cross-section at the default meteorology is available at <a href="https://doi.org/10.5281/zenodo.14709364">https://doi.org/10.5281/zenodo.14709364</a>

#### **Author contribution**

CAM modified the CoCiP code, wrote the post-processing code and performed the simulations. SE helped with the HPC setup and provided technical guidance. JPJ set the overall scope of the manuscript, and the structure of the experimental design. CAM prepared the manuscript with reviews and edits from all co-authors.

#### 575 Competing interests

The authors declare that they have no conflict of interest.

#### Acknowledgements



The authors would like to thank Dr. Jessie R. Smith for reviewing early versions of this work. The authors would also like to thank Paul Bond for his help setting up APCEMM initially, and to Michael Xu for the help with the APCEMM troubleshooting. The authors would also like to thank Tristan Abbott for his assistance in understanding the cause of the outlier CoCiP behavior associated with 125 % layer RHi. This work was supported by the UK Engineering and Physical Sciences Research Council [grant number EP/W524633/1]. The authors recognize funding used for the development of APCEMM from NASA. Development of the APCEMM model used in this study by co-author Eastham was partially supported by the U.S. National Aeronautics and Space Administration through grant NNH20ZEA001N. For the purpose of open access, the authors have applied a Creative Commons Attribution (CC BY) license to any Author Accepted Manuscript version arising. This work was performed using resources provided by the Cambridge Service for Data Driven Discovery (CSD3) operated by the University of Cambridge Research Computing Service (www.csd3.cam.ac.uk), provided by Dell

EMC and Intel using Tier-2 funding from the Engineering and Physical Sciences Research Council (capital grant EP/T022159/1), and DiRAC funding from the Science and Technology Facilities Council (www.dirac.ac.uk).

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
