# Peer review of "Zero-dimensional contrail models could underpredict lifetime optical depth"

_EGUsphere, 2025_

## Referee Comment (RC2)

**Contrail models lacking post-fallstreak behavior could underpredict lifetime optical depth**

Report of Reviewer #3

**Summary**

The study presents a timely analysis comparing two widely applied simplified contrail models that are typically used to evaluate contrail mitigation options. The benefit of mitigation options in terms of reducing some contrail climate metric can in most cases only be based on model-based estimates. Rerouting strategies aim to avoid 'big-hit' contrails. Yet, the prediction of those high-impact contrails with simplified models is challenging. The skill of the models is unknown as the validation with observations is limited in the sense that such extreme contrails have not been sampled and is unlikely to be possible due to their long lifetime. Hence, comparing two models for a set of meteorological parameter combinations is very interesting and the study is insightful.

**General comments**

1. The writing style is in my opinion too informal and formulations are too often too vague and not precise enough. Moreover, it is not explained sufficiently how you compute specific values.
   Some examples are listed in the specific and technical comments, but my list will not be exhaustive. Hence, I strongly recommend that the authors go over the whole manuscript and work on the text. Sloppy formulations make the life of the reviewers and future readers harder than it should be.

2. I am not sure if "fallstreak" and "post-fallstreak" are good expressions for what you want to describe. The first phase (that you refer to as "fallstreak") is dominated by the creation of the fallstreak that fills the moist layer underneath the flight altitude over time. Once the fallstreak covers the whole layer, you speak of "post-fallstreak". In my understanding, the contrail at that stage still consists of a contrail core and a fallstreak. The fallstreak continues to exist and is fed by ice crystals falling out of the contrail core. Hence, I would not call it "post-fallstreak".
   [In the following review, I will stick to your terminology and will not make any further comments whether I think the terminology is appropriate.]
   Moreover, you speak of a Cocip fallstreak. As a single Gaussian plume is used in Cocip, this model cannot represent the bimodality of the contrail (i.e. contrail core and fall streak as e.g. described in the high-resolution modelling study by Lewellen 2014). Hence, referring to the Cocip plume as Cocip fallstreak is misleading as the Cocip plume falls only slowly in the beginning and accelerates only very late in its lifecycle. Why not use the more neutral term 'Cocip (Gaussian) plume or contrail' throughout the text?

3. The prescribed meteorological scenarios are highly idealized and it is likely that subsidence causes the contrail to sublimate before it reaches an age of 15 hours. Hence, the comparison should emphasize the early differences more than the discrepancies beyond 10 hours. I doubt the "fading sub-regime" will be encountered as such very often.
   The change in the slope of the ice crystal number reduction might be a particular

result of APCEMM and the idealized scenario used. In reality, vertical motions in the atmosphere will perturb the contrail evolution.

**Specific comments**

4.  Line 39: Can you substantiate the statement about thin contrails having the largest cloud radiative effect (what you call local RF)? Unterstrasser & Gierens (2010) and Lewellen (2014) show at least the dependence on wind shear.
5.  Line 40: Typically, the introduction of scientific publications does not summarize the results of the present study.
6.  Line 44: In my opinion, contrails with cross-sections of $100 km^2$ represent extreme cases. Or did you want to say, the grid boxes of the gridded models are $100 km^2$? In this case, please reformulate.
7.  Line 47: This sentence is very general and contains little information. Which models were compared to each other? In which paper was the comparison done? What are the main findings?
8.  Line 77: I thought Cocip only tracks a Gaussian plume for the ice crystals and the humidity is taken from NWP data. Why is it necessary to have a plume of water vapour concentrations?
9.  Line 96: are the bins fixed in radius space or dynamic as in Lewellen 2014?
10. Line 82: 'evaporate' or is it 'vanish/disappear'?
11. Line 114: I am not sure whether the title is appropriate. Could you reformulate it? "Meteorological background scenarios/data"??
12. The quantities you define in Eqs. 1 and 2 have been used in previous studies, yet with other names. It would be good to make the connection to those studies. Unterstrasser & Gierens 2010 introduced the total extinction, which is equal to your definition of γ. Since then, total extinction has also been evaluated in the context of GCM contrail simulations (Bier et al, 2017). Moreover, total surface area S in Lewellen 2014 is basically the same as total extinction (except for a constant scaling factor of 2).
    Your definition of "lifetime optical depth" was introduced as '(life)time-integrated total extinction' in Unterstrasser (2020).
    I would recommend to stick to one of the names that have been previously introduced to make clearer that all these studies analyse basically the same quantity.
13. The crystal loss rate is not well-introduced and I stumble across the units. Is the logarithmic derivative of N(t) used?
14. Around line 190: You analyse $dI/dt$ and $d^2I/dt^2$ which serve as conditions in a contrail phase classification. It makes the impression that those conditions can used as classification criterion across different scenarios. I doubt that the signs of these two quantities are universally interpretable as they may depend on many parameters (such as the thickness of the moist layer, vertical air motions and so forth).
15. Section 3.2.1: The study by Bier et al (2017) also analysed what factors limit the contrail lifetime.
    Matching maximum ISSR lifetimes alone are not a sufficient criterion ensuring that your meteorological background state is representative. The characterization of the sub-regimes is more complicated in scenarios where the background humidity changes over time due vertical air motions and ice mass evolution changes by

those 'external' drivers. Hence, your claim of widespread applicability is probably a bit overselling.

16. If I understand Fig. 6a correctly, the x-coordinate for the blue and the according orange data point are the same. Correct?

    I understand the information given in the text about which fraction of the contrail lifecycle is unobservable (based on τ<0.1). Basically, Cocip contrails are nearly always observable and 100% of their lifecycles belong to the fallstreak regime. Due to these rather peculiar values, the panel b is difficult to understand. First of all, the legends in the two panels say 'fallstreak only' and 'fall streak'. Is this the same criterion?

    In the text you mention that on an aggregate level, 92% of $\Gamma_{APCEMM}$ comes from post-fall streak regime. It is not explained how you derive this number. Is this the ratio of the orange and blue slope in Fig. 6a? Are all data points equally weighted in the averaging? Do you take the average over the ratios $\Gamma_{APCEMM, fallstreak}$ / $\Gamma_{APCEMM all}$ ? Or do you sum up over $\Gamma_{APCEMM, fallstreak}$ and $\Gamma_{APCEMM all}$ separately and then compute the ratio of the two sums?

    Similarly, I miss information about how the values 35% and 15% (in lines 248 and 249) are computed. Are these the mean values of the orange data points in x and y direction in Fig. 6.2?

17. section 4.2.2: You compare APCEMM and Cocip sensitivities with those found in Lewellen 2014. Unterstrasser & Gierens 210a,b also studied contrails in scenarios with constant RHi and analysed the sensitivities to most of the parameters listed in your table 2. Hence, it would help including the findings from these studies in your discussion.

**Technical comments**

18. Abstract, first line: what does 'optimized' imply?
19. Line 56: sublimation is a specific physical process, whereas formation and persistence are more general terms. Moreover, contrails can disappear through other physical processes. Replace 'sublimation' by 'demise'?
20. Fig.2: Would it be possible to use white as colour for the zero IWC bin, which would help to better identify the borders of the contrail?
21. In Eq.3, should the index be 3 (and not 'n')?
22. Line 199: "During the fall streak"?? and "fastest center of mass fall rate" (a rate is not fast, it is large)
23. Line 214: I am not sure what **local** optical depth means.
24. Line 215: 'is produced at **times** where …'. Better use time instead of point to make clear it is about time and not space. Moreover, I would prefer to use plural to make clear you consider a time span over which the contrail is not detectable.
25. Caption Fig.6: 'unobservable'
26. Line 262: 'at the end of the APCEMM fallstreak **regime(?)'; '**shear does not increase the contrail width'. It is true that shear increases the contrail width. But here you want to say that a larger shear value leads to a larger contrail width.
27. Line 337: 'once the fallstreak ends': in time or space?
28. Line 345 I believe it should be 'Contrail avoidance strategies **that**' because the following clause is restrictive. Same in line 385: 'which' -> 'that'.
29. Line 348: and also lifetime-integrated radiative effects?

**References**

[Only those are listed that are not listed in the manuscript under review.]

Bier, A., Burkhardt, U., and Bock, L.: Synoptic Control of Contrail Cirrus Life Cycles and Their Modification Due to Reduced Soot Number Emissions, J. Geophys. Res., pp. 11 584–11 603, https://doi.org/10.1002/2017JD027011, 2017JD027011, 2017.

Unterstrasser, S.: The Contrail Mitigation Potential of Aircraft Formation Flight Derived from High-Resolution Simulations, Aerospace, 7, 170, https://doi.org/10.3390/aerospace7120170, 2020.

---

## Author Comment (AC1)

**Responses to Anonymous Referee #1**

**"I do not agree with the final conclusion that it would be currently better to avoid all contrails instead only the ones with strongest warming. With the current ATC system this would not be possible, at least not in dense airspaces."**

**Author's response:** We agree - our intention is to highlight that there is insufficient agreement regarding which contrails are "big hits" between models, and thus more robust strategies should be considered. An example of a robust strategy is unbiased avoidance. This is because avoiding a certain proportion – say, 5% - of persistent contrails (with equal probability of targeting each individual persistent contrails), will result in a proportional reduction in contrail warming. In theory one could use a contrail model to instead select the most impactful 5%, but our results indicate that the ones chosen are likely to differ depending on the model – and that in fact one model may prioritize contrails which another model specifically predicts to be the least impactful. We have modified the manuscript throughout to more clearly reflect this argument.

**Author's changes in manuscript:** Clarified in the conclusions, as well as throughout the manuscript, that the strategy that we propose is unbiased avoidance.

**"My question here is whether it is actually monodisperse which means that every ice crystal has the same size (variance zero), or whether it rather may have a variance which is implicit. Note that many bulk cloud physics models (1-moment or 2-moment) implicitly assume a size distribution, where the variance is a function of the mean size. Please check, how CoCiP does this, i.e. whether it is monodisperse or perhaps monomodal.**

**Later, you say that CoCiP uses the ice mass and number and that the single size of all crystals is computed from these two values. There may be reasons for this, but in cirrus models, if they have 2-moment schemes (e.g. ice mass and number), one of the reasons for this is that this allows more freedom in treating size distributions implicitly."**

**Author's response:** We agree that the description of the way CoCiP handles the different crystal sizes was not sufficiently clear in the manuscript. In short, CoCiP, tracks the total ice mass and total crystal number only, with various different implicit assumptions made regarding the size distribution. The pycontrails implementation of CoCiP makes the following implicit assumptions about the ice radius distribution:

- The fall speed is calculated using a parametric model from Section 3.1.3 in Spichtinger and Gierens (2009). This calculates a single fall speed for all crystals.
- For the optical depth, extinction is calculated based on a single radius (line 1020 in contrail_properties.py from the pycontrails CoCiP implementation).
- CoCiP attempts to compensate for the lack of gravitational sorting by enhancing its vertical diffusivity, but still does not explicitly represent the particle size distribution (Schumann, 2012). See Equation 25 of Schumann (2012) for further information.

- For the ice particle size, CoCiP tracks the ice mass and the number of ice crystals separately. The ice mass is calculated based on the following statement "The water mass MH2O in the segment is composed of contrail water in the ice phase and water in the vapor phase at ice-saturation" (Schumann, 2012). Therefore, the ice mass is the mass of the water vapor minus the mass of water vapor at saturation at each timestep (see equation 44 in Schumann, 2012).
- The only equation that makes an assumption about the particle size distribution (PSD) is the equation estimating ice number loss due to aggregation. Schumann (2012) says "Since we have no explicit information on the particle size spectrum, we assume that the size spectrum in the contrails has a width of order r ".

With this evidence, we can conclude that indeed the CoCiP model is monodisperse in most relevant cases. The only place in which a non-monodisperse PSD is assumed is in the ice number loss due to aggregation. We have added this exception to the manuscript to make it explicitly clear.

Spichtinger, P. and Gierens, K. M.: Modelling of cirrus clouds – Part 1a: Model description and validation, Atmos. Chem. Phys., 9, 685–706, https://doi.org/10.5194/acp-9-685-2009, 2009.

**Author's changes in manuscript:** Added the following in line 108 of the revised manuscript:

"The monodisperse assumption is used when calculating the ice particle size, fall speed, and optical depth of the Gaussian plume. However, ice crystal loss due to aggregation is also modelled in CoCiP (see Eq. 52 in Schumann, 2012), for which the width of the size spectrum is assumed to be of order r (Schumann, 2012). Although the crystal loss parameterizations implicitly assume a size distribution, all crystals are treated identically with a single radius value and no size distribution is diagnosed. As such, we still refer to CoCiP as a monodisperse model."

**"Figure 5 left shows that dN/dt = -N/τ, that is the crystal loss rate is not constant. Wouldn't a monodisperse distribution of crystals, falling with identical speed, lead to a constant loss rate? Furthermore, if all crystals do what the crystal in the contrail centre does, why then is there ongoing crystal loss instead of instantaneous vanishing of all crystals?"**

**Author's response:** This is true. Following the initial loss of ice crystals during vortex sinking, CoCiP assumes continuous loss of ice crystals through three mechanisms – parameterizing some of the effects of a non-monodisperse size distribution to estimate the effect on the two stored values (total ice mass and total crystal number):

- Losses due to internal plume turbulence (denoted as "turb" in Equation 49 from Schumann (2012)):

$$\text{``}(dN/dt)_{turb} = -E_T \left( \frac{D_H}{\max(B,D)^2} + \frac{D_V}{D_{eff}^2} \right) N \text{''}$$

- Losses due to sedimentation-induced aggregation (denoted as "agg" in Equation 52 from Schumann (2012)):

$$\text{``}(dN/dt)_{agg} = -E_A 8\pi r^2 V_T N^2/A\text{''}$$

- Losses by turbulent humidity fluctuations, mesoscale turbulence, and gravity waves (denoted as "meso" in Equation 55 from Schumann (2012))

$$\text{``}(dN/dt)_{meso} = -E_{meso} N_{BV} w'_{meso} (dT/dz)/\Delta T_c\text{''}$$

The only other mechanism for loss of crystals is total evaporation of the contrail. Once subsaturated air begins to mix with the contrail, all crystals will give up ice to maintain 100% saturation – meaning that the ice mass (and therefore the effective radius) decreases uniformly. Once one of the end-of-life conditions is reached, all crystals are eliminated instantaneously. This can take a few time steps but is typically very rapid as CoCiP does not model the horizontal distribution of water vapor or crystals within the contrail air mass.

The equations for these losses depend on several time-varying contrail and ambient properties. Hence, even though CoCiP is a monodisperse model, the ice loss rate is not constant when the plume is in supersaturated air.

We have updated the manuscript to explain why the ice loss rate is not constant in an ISSR, and why contrail demise is not instantaneous when leaving the moist layer.

**Author's changes in manuscript:** Added a new appendix - "Appendix H" containing the above explanation.

**"In line 200 you write "average ice particle". If the distribution is monodisperse, all ice crystals are average."**

**Author's response:** This is correct. We have modified the manuscript accordingly.

**Author's changes in manuscript:** Replaced "with the fall speed of the average ice particle" with "due to its monodisperse ice radius distribution" in line 234 of the revised manuscript.

**"Section 2.2, Eqs 1 and 2: Why is the spatial integral only over the width of a contrail and not along its length? This quantity as used here seems to have some similarity to the "total extinction" of Unterstrasser and Gierens and a similar quantity introduced by Lewellen. These authors use them as proxies for climate impact, perhaps an instantaneous one. But in the present case, I fear this could not serve the intended comparison. The total radiative effect of a contrail should be the lifetime integral of the vertical optical thickness at every point of the contrail (width X length). For the current purpose, length should somehow increase with lifetime, and it seems that this effect is overlooked. Wouldn't the differences between CoCiP and APCEMM be even larger if the integrals would cover the complete contrail area over the complete lifetime?"**

**Author's response:** First, we agree that the "total extinction" metric is a more appropriate one and thank the reviewer for this suggestion. We have renamed our "lifetime optical depth" variable to instead be the "time-integrated total extinction" to maintain consistency across different works.

Second, we chose to only consider the evolution of the contrail in the 2D sense to ensure we capture the native model behavior for APCEMM and CoCiP, without any addons to represent the entire flight-wise span of the contrails. This is because, although the models can be programmed to axially stretch the contrail in a consistent way, this would not change the conclusions drawn regarding the behavior of the contrails in the absence of axial stretching. We believe that the "time-integrated total extinction" is an appropriate metric for this purpose. We acknowledge that the manuscript was originally unclear in this regard and have added a clarifying sentence in the methodology.

**Author's changes in manuscript:** Replaced "integrated optical depth" with "total extinction", and "lifetime optical depth" with "time-integrated total extinction" throughout the manuscript. Added clarifying statements in the methodology to further justify the chosen impact metric and scope of the manuscript: "and no effects in the flight direction are considered" (in line 152 of the revised manuscript) and "Given that we do not consider effects along the flight direction, the time-integrated total extinction accounts for persistence, lateral spread, and optical properties" (in line 167 of the revised manuscript).

**"The title can be improved. It is not clear what "lifetime optical depth" may be. I think, the problem is not the optical depth, but the lifetime-integrated radiation effects or the change of the radiation energy flow integrated over the contrail lifetime.**

**When the same expression appears in the abstract, it should be written as "lifetime-integrated optical depth". As an explanation is given (proxy), the expression is acceptable, but in the title it should be changed. At the end of the abstract you explicitly write "contrail climate impact", why not so in the title?"**

**Author's response:** We appreciate the comment and share the opinion that our original choice of language may not be optimal. In response to this, the authors had a meeting to discuss alternative titles, but we found that more specific language took away from the effectiveness and conciseness found in the original. For that reason, we have chosen to abstain from changing the title. We do recognize, however, that the terminology used in the abstract is mistaken, and have made the appropriate changes.

**Author's changes in manuscript:** Changed "contrail climate impact" in the abstract to "the time-integrated total extinction" (line 10 of the revised manuscript).

**"Abstract, line 19: "a strategy avoiding all contrail formation is still expected to yield a reduction in climate impact". I believe such a strategy does not work for practical reasons (ATC problems) and does not exist therefore. I also do not believe that such a strategy has been proposed, as written in Line 25."**

**Author's response:** .We agree that our original presentation was inaccurate given the point that we are trying to make. As stated in comment RC1-01, we believe that unbiased avoidance is more robust than strategies using contrail models to select the contrails to avoid, provided that the same proportion of flight length is avoided, irrespective of what that proportion is. We have modified the manuscript to clarify our point. Secondly, we appreciate the comment regarding the use of the word "proposed" in line 25. We agree that this word is misused and unnecessary. We have therefore chosen to remove it.

**Author's changes in manuscript:** Removed the word "proposed" in line 25 of the original manuscript.

Replaced: "While a strategy avoiding all contrail formation is still expected to yield a reduction in climate impact, implementing optimized requires more research to establish confidence in model predictions"

with: "While a strategy avoiding a given proportion of persistent contrails in an unbiased way is still expected to yield a proportional reduction in the time-integrated total extinction, implementing strategies using contrail models to select the specific contrails to avoid may lead to fewer reductions in the time-integrated total extinction, primarily due to the current level of disagreement between models. We therefore recommend more research to establish confidence in model predictions at later contrail ages" in line 19 of the revised manuscript (part of the abstract).

**"Line 29: why hypothetical? Why not as well in actually occurring situations?"**

**Author's response:** We agree with the point that prediction use cases for contrail models are not just limited to hypothetical scenarios. We have adjusted the manuscript accordingly and are grateful for this comment.

**Author's changes in manuscript:** We replaced "rely on contrail models to accurately predict net contrail warming in hypothetical scenarios" with "rely on the availability of accurate contrail models" in line 32 of the revised manuscript.

**"Line 44: cross sectional area of ~ 100 km$^2$, what kind of cross section do you mean? Say a contrail is 1 km deep, than it must be 100 km broad in your example."**

**Author's response:** We have added a clarifying statement, with reference to previous studies, to address this.

**Author's changes in manuscript:** Replaced: "since they have cross sectional areas of ~100 km$^2$"

with: "A study by Dickson et al. (2009) found that 53 % of the ISSRs they observed were between 100 and 1500 m deep, and the large eddy simulations conducted in Lewellen (2014) had widths of ~50 km in the transversal direction (defined to be along the horizontal plane perpendicular to the flight direction). Furthermore, a single flight was identified as responsible for creating a cirrus cloud with a bounding box width of 130 km (measured from Fig. 12 (c) in the study by Haywood et al. (2009)). Therefore, the largest persistent contrails can reach cross-sections of up to ~100 km$^2$ in the transversal direction, making gridded simulations of sufficient resolution computationally expensive" in line 47 of the revised manuscript

Haywood, J. M., Allan, R. P., Bornemann, J., Forster, P. M., Francis, P. N., Milton, S., Rädel, G., Rap, A., Shine, K. P., and Thorpe, R.: A case study of the radiative forcing of persistent contrails evolving into contrail-induced cirrus, J. Geophys. Res.-Atmos., 114, https://doi.org/10.1029/2009JD012650, 2009.

**"Line 47: "The limited comparisons that have already been performed for large eddy simulations indicate disagreement in this regard", can you be more specific? As far as I remember, the cited papers weren't model comparison papers. What do you mean?"**

**Author's response:** We agree that the original sentence was vague and did not provide much insight. We have now modified the manuscript to include some points of agreement and disagreement between Lewellen (2014) and Unterstrasser and Gierens (2010a), and Unterstrasser and Gierens (2010b). According to Lewellen (2014), "some of the inferences given in UG10a and UG10b are not supported by the present study" and "several of the parameter dependencies discussed here were found previously in UG10a and UG10b". In particular, both studies determined that the total extinction increases with the relative humidity, temperature, and initial contrail ice number. However, they found different parameters dominating the changes in total extinction: relative humidity in Unterstrasser and Gierens (2010a and 2010b), and shear in Lewellen et al. (2014) and Lewellen (2014).

**Author's changes in manuscript:**

Replaced "The limited comparisons that have already been performed for large eddy simulations indicate disagreement in this regard (Unterstrasser and Gierens, 2010a; Unterstrasser and Gierens, 2010b; Lewellen et al., 2014; Lewellen, 2014)"

With "A study employing full-lifetime large eddy simulations (Lewellen, 2014) compared its findings with a prior similar study (Unterstrasser and Gierens, 2010a – UG10a; Unterstrasser and Gierens, 2010b – UG10b), and found that "some of the inferences given in UG10a and UG10b are not supported by the present study" and "several of the parameter dependencies discussed here were found previously in UG10a and UG10b" (Lewellen, 2014). Specifically, both studies determined that the total extinction (a proxy climate impact metric) increases with the relative humidity, temperature, and initial contrail ice number. However, they found different

parameters dominating the changes in total extinction: relative humidity in Unterstrasser and Gierens (2010a and 2010b), and shear in Lewellen et al. (2014) and Lewellen (2014). Since two models of similar complexity found different dominant factors in predicting a proxy for contrail climate impact, this suggests the need for a more comprehensive assessment of the robustness of contrail modelling techniques being used to inform contrail impact mitigation" in line 56 of the revised manuscript.

**"Line 50: "inconsistencies"? Probably you simply mean model differences or disagreements or contradictory results. To my view, two different models, independently developed, can neither be consistent nor inconsistent."**

**Author's response:** We agree with the comment and are grateful for it. We think that the best way to describe this is "differences between the behavior of the models"

**Author's changes in manuscript:** We replaced "inconsistencies between the models" with "differences between the behavior of the models" in line 67 of the revised manuscript.

**"Lines 68-70: The two sentences "Teoh shows…" and "For this reason" are not logically connected, to my opinion. To only consider long-lasting contrails is justified without Teoh's results. Whether the latter turn out tenable can be doubted in view of your results, in particular the probably wrong sensitivity to layer supersaturation lets me doubt to which degree Teoh's results are believable."**

**Author's response:** We are grateful for bringing this non-sequitur to our attention. To address this, we added a sentence that bridges the jump in logic. Regarding the validity of the study conducted by Teoh et al. (2024), we would like to emphasize that no one model is "right", or "better" than another since we recognize that there is no ground truth covering cases at such late lifetimes. We are therefore unable to comment on the validity of the results presented by Teoh et al. (2024).

**Author's changes in manuscript:** Added "This implies that the most warming contrails produce the majority of their climate impact in the diffusion regime" in line 89 of the revised manuscript.

**"Line 109: "Equivalent"? Is there a subtle meaning that I do not understand or do you just mean "Equal"?"**

**Author's response:** Although this comment raises a good question over our use of the word "Equivalent", it reveals some underlying ambiguity in the sentence. The meteorological scenarios are identical but the meteorological inputs are not exactly identical due to differences in the required input format for each model. To address this, we have clarified the entire sentence.

**Author's changes in manuscript:** We replaced the following sentence in line 135 of the revised manuscript: "Equivalent meteorological scenarios are then produced for each model, with each scenario described by six independent meteorological parameters (Section 2.1)" with "Meteorological inputs for each model are then produced for each simulation scenario, each of which described by six independent meteorological parameters (Section 2.1)."

**"Line 216: correct "observable in from satellites"."**

**Author's response:** This has been corrected.

**Author's changes in manuscript:** We replaced "observable in from" with "observable from" in line 251 of the revised manuscript.

**"Figure 6: The figures are not entirely understandable. Partly, because they are just scatter plots and it is not clear which CoCiP point is paired to which APCEMM cross. Then, while CoCiP should indeed be represented by a single group of points for "fallstreak only", there should be 2 groups of crosses for APCEMM: "fallstreak only" and "all phases". I suggest, to have the integrated optical thickness on the y-axis, while the x-axis should be numbered 1-14, that is the number of the sensitivity experiment. Then each number would have one blue point for CoCip, and, say a red point and a red cross for APCEMM "fallstreak only" and "all phases". A similar outline would also work for the rhs panel."**

**Author's response:** We extend our appreciation for the thoughtful suggestions made in this comment. Upon review, we have concluded that the caption and legend of this figure are not clear enough for most readers to interpret these properly. Nevertheless, we have decided to keep the original Figure 6 format while modifying the caption and the legend to clarify how the data should be interpreted. We believe that the parity plot format adds an additional layer of insight through the visualization of the agreement of the entire simulation suite. This capability is not present in many other visualization techniques. However, if this Figure is raised again as a source of confusion, we are happy to replace it with an alternative. Furthermore, we would like to inform Referee 1 that we have updated the terminology used to describe the sub-regimes. More details can be found in RC3-02.

**Author's changes in manuscript:** Added the following sentence to the caption of Figure 6(a): "Each entry in the parity plot corresponds to one simulation. In (a), the crosses indicate the

simulations where the whole lifetime has been considered, whereas the dots indicate the simulations where only the unrestrained sub-regime has been considered."

**"Figure 6, caption: correct "unobserbavle"."**

**Author's response:** Fixed.

**Author's changes in manuscript:** Changed "unobserbavle" to "unobservable" in the caption for Figure 6.

**"Line 300: "Varying the layer RHi causes the lifetime optical depth to decrease in CoCiP and increase in APCEMM". This sentence is a bit unclear, since the word "varying" includes both decreasing and increasing. Please correct."**

**Author's response:** We are grateful for highlighting this ambiguity. We have now resolved it.

**Author's changes in manuscript:** Replaced the word "Varying" with "Increasing" in line 367 of the revised manuscript.

**"Line 345/6: The Schmidt-Appleman criterion says nothing on contrail persistence, so I suggest to add ice supersaturation as a condition. Avoidance of all, that is, including very short contrails, is not useful and probably worsening the climate (unnecessary fuel consumption and emissions)."**

**Author's response:** This comment highlights an oversight in the manuscript: this sentence should have considered persistent contrails only. We have corrected the sentence.

**Author's changes in manuscript:** Specified that avoiding all persistent contrails can be done by predicting the Schmidt-Appleman criterion and ice supersaturation in lines 416-418, as well as in line 31 of the revised manuscript.

**"Line 385: "our results suggest that contrail avoidance strategies which focus on avoidance of all contrails will have the greatest chance of producing a real climate benefit". I would not subscribe to this conclusion. It renders contrail avoidance practically**

**impossible, in particular for ATC reasons. The ATC sectors where no contrails form would become overcrowded if they are neighbours to sectors where contrails can form. I think, the appropriate conclusion of your test is that more work is needed to "calibrate" the models to realistic behaviour and to test whether the promised results are satisfying. Your Section 5 points to this direction and I fully agree to the statements of Sect. 5."**

**Author's response:** This was a miscommunication – we are not proposing avoiding all persistent contrails. Instead, we propose avoiding contrails over a certain flight proportion in an unbiased fashion, as opposed to model-informed contrail avoidance. Please refer to our response to comment RC1-01 for more details.

**Author's changes in manuscript:**

Replaced "our results suggest that contrail avoidance strategies which focus on avoidance of all contrails will have the greatest chance of producing a real climate benefit" with "our results suggest that unbiased contrail avoidance strategies at any scale will have the greatest chance of producing a real climate benefit" in line 461 of the revised manuscript.

---

## Author Comment (AC2)

**Responses to Anonymous Referee #3**

**"The writing style is in my opinion too informal and formulations are too often too vague and not precise enough. Moreover, it is not explained sufficiently how you compute specific values. Some examples are listed in the specific and technical comments, but my list will not be exhaustive. Hence, I strongly recommend that the authors go over the whole manuscript and work on the text. Sloppy formulations make the life of the reviewers and future readers harder than it should be."**

**Author's response:** We appreciate the comment and have gone through the text as suggested, correcting not only the specific issues highlighted by the reviewer but also seeking to be more specific with our technical explanations. We apologize for any difficulties caused by our oversights, and hope that the modifications made to the manuscript have resolved this issue.

**Author's changes in manuscript:** Several changes have been made throughout the text, improving formulations, choice and clarity of language, and adding units where they were lacking.

**"I am not sure if "fallstreak" and "post-fallstreak" are good expressions for what you want to describe. The first phase (that you refer to as "fallstreak") is dominated by the creation of the fallstreak that fills the moist layer underneath the flight altitude over time. Once the fallstreak covers the whole layer, you speak of "post-fallstreak". In my understanding, the contrail at that stage still consists of a contrail core and a fallstreak. The fallstreak continues to exist and is fed by ice crystals falling out of the contrail core. Hence, I would not call it "post-fallstreak".**

**[In the following review, I will stick to your terminology and will not make any further comments whether I think the terminology is appropriate.]**

**Moreover, you speak of a Cocip fallstreak. As a single Gaussian plume is used in Cocip, this model cannot represent the bimodality of the contrail (i.e. contrail core and fall streak as e.g. described in the high-resolution modelling study by Lewellen 2014). Hence, referring to the Cocip plume as Cocip fallstreak is misleading as the Cocip plume falls only slowly in the beginning and accelerates only very late in its lifecycle. Why not use the more neutral term 'Cocip (Gaussian) plume or contrail' throughout the text?"**

**Author's response:**

We appreciate that our choice for the sub-regimes identified in this study are not the most appropriate given the pre-existing connotations of the word "fallstreak". After much consideration, we have come up with these alternatives:

- Fallstreak sub-regime -> unrestrained sub-regime
    - We believe that the word "unrestrained" accurately reflects the fast vertical separation of the precipitation plume from the contrail core.

- Settling sub-regime -> restrained sub-regime
    - The  contrail now spreads in a slower manner and primarily in a horizontal direction, so we believe that "restrained" is more appropriate than "settling".
- Fading sub-regime -> unchanged

Furthermore, we have replaced the reference to "fallstreak" in the title with "Zero-Dimensional model" to avoid confusion.

**Author's changes in manuscript:** Changed the sub-regime names in the manuscript text and figures as required. Also changed the title from "Contrail models lacking post-fallstreak behavior could underpredict lifetime optical depth" to "Zero-dimensional contrail models could underpredict lifetime optical depth"

**"The prescribed meteorological scenarios are highly idealized and it is likely that subsidence causes the contrail to sublimate before it reaches an age of 15 hours. Hence, the comparison should emphasize the early differences more than the discrepancies beyond 10 hours. I doubt the "fading sub-regime" will be encountered as such very often.**

**The change in the slope of the ice crystal number reduction might be a particular result of APCEMM and the idealized scenario used. In reality, vertical motions in the atmosphere will perturb the contrail evolution."**

**Author's response:** This comment raises two points. First, it suggests that we should focus on the contrail simulations up to 10 hours due to how uncommon contrails lasting beyond this are. To consider this suggestion in a fair manner, we have produced an additional figure showing the sum across all unique simulations of the difference in time-integrated total extinction between the models. For ease of interpretation, this has been normalized with the total difference (summed across all simulations) at 24 hours. Each time on the x axis serves as the upper limit of integration, hence the variation with time.

[Figure]

After much consideration, we have decided to not shift the focus of the analysis to the first 10 hours. This is because, according to the figure above, 90 % of the model difference in the time-integrated total extinction is produced within 12 hours from formation. Hence, our conclusions

still hold even if contrails that persist for longer 12 hours are rare. Nevertheless, we are grateful for this suggestion for the opportunity it gives us to improve the narrative. We have included this figure and the above explanation in Section 3.2.1.

The second point raised by this comment is later reiterated in RC3-14. In summary, we agree that our mathematical definitions of the sub-regimes may not hold in all cases. However, we provide these definitions as a way to deepen our understanding of the behavior of contrails. Please refer to the response to RC3-14 for a more detailed response.

**Author's changes in manuscript:** Included the above figure and explanation in a new section: Section 4.1.3:

"To understand the sensitivity of our findings to the contrail lifetime, we define the global model difference ($\delta$) as the sum across all simulations of the APCEMM integrated total extinction minus the CoCiP integrated total extinction at each timestep:

$$\delta(t) = \sum_{\text{all cases}} \left( \hat{E}_{\text{APCEMM}}(t) - \hat{E}_{\text{CoCiP}}(t) \right), \tag{5}$$

$$\hat{\delta}(t) = \frac{\delta(t)}{\delta(t = 24\,\text{h})}. \tag{6}$$

where $\hat{\delta}(t)$ is the normalized global model difference. The variable t in Eqs. 5 and 6, is the upper limit of integration in Eq. 2.

Figure 10 shows how $\hat{\delta}(t)$ varies as a function of time. We hence find that 90 % of the global model difference is produced within 12 hours from formation. For more evidence-based contrail lifetime estimates, we take 4 h and 8 h from a recent preprint by Hofer and Gierens (2025). The proportion of the total model difference reached by 4 h and 8 h are 27 % and 72 % respectively. These results indicate a large sensitivity in our findings to the lifetime of typical contrails. However, they also indicate that our findings are particularly relevant to those 6–7 % of contrails that persist beyond 8 h (Gierens and Vazquez-Navarro, 2018). Such contrails are also likely to be the greatest contributors to aviation warming on an individual basis, and are hence important for contrail avoidance."

For the resolution to the second point raised in this comment, please refer to the changes listed in a later comment.

**"Line 39: Can you substantiate the statement about thin contrails having the largest cloud radiative effect (what you call local RF)? Unterstrasser & Gierens (2010) and Lewellen (2014) show at least the dependence on wind shear."**

**Author's response:** We appreciate that we have been vague in this explanation, and we welcome the suggestion to substantiate the point. Please find here an explanation of what we mean:

For the following analysis we reproduce part of the curve corresponding to 5 µm from Fig. 4a) of Wolf et al. (2023):

[Figure]

Consider two contrail segments under the same ambient conditions with the same ice mass per unit contrail length of 20 kg m-1, with both contrails having the same depth of 500 m. Assuming that contrail A is 1 km wide, and that contrail B is 2 km wide, the ice water content (IWC) of contrail A ($0.04$ g m$^{-3}$) is two times the IWC of contrail B ($0.02$ g m$^{-3}$). Fig. 4b) of Wolf et al. (2023) shows that contrail A will have an instantaneous longwave radiative forcing (LW RF) of ~125 W m-2, while contrail B will have an instantaneous LW of ~115 W m$^{-2}$. Accounting for the contrail width, contrail A will have a LW RF of ~125 W m$^{-1}$, whereas contrail B will have a LW RF of ~230 W m$^{-1}$. Contrail A, the one with higher optical depth (due to its higher IWC), will have a lower energy forcing than the more dilute but wider contrail B. This implies that, for the same total ice mass, contrails that have a large horizontal span but are optically thin may have a greater climate impact than thicker, narrower contrails.

**Author's changes in manuscript:** Created a new Appendix "G" with the above explanation. In line 41 of the updated manuscript, we replaced:

"Since the local RF of an ice cloud increases fastest with ice water content for thin contrails"

with "Using data from Fig. 4(a) from Wolf et al. (2023) it can be shown that, out of two contrails of different width but the same length, depth and total ice mass, the wider contrail has a higher energy forcing than the narrower contrail (see Appendix G)." and removed "such large, thin contrails are expected to have greater total climate impact than narrower, more visible contrails".

**"Line 40: Typically, the introduction of scientific publications does not summarize the results of the present study."**

**Author's response:** We agree that the quantitative discussion of a simulation is not appropriate for the introduction, and have removed explicit discussion of the results in the introduction. Although Figure 1 does show a result of sorts, we have decided to keep it as we believe that it works well as an illustrative example.

**Author's changes in manuscript:** Replaced "For a typical contrail simulated in APCEMM (Fig. 1), assuming an optical depth observability threshold of 0.1 (Kärcher et al., 2009), 25 % of the lifetime optical depth is produced in the unobservable period"

with "It is hence possible that a significant proportion of the time-integrated total extinction (a proxy for climate impact) remains unaccounted for when relying on observations, as shown by the illustrative example in Fig. 1" in line 45 of the updated manuscript

**"Line 44: In my opinion, contrails with cross-sections of 100km2 represent extreme cases. Or did you want to say, the grid boxes of the gridded models are 100km2? In this case, please reformulate."**

**Author's response:** We agree that the chosen value of 100km$^2$ is representative of extreme cases only and have amended the text to indicate a way this estimate is calculated

**Author's changes in manuscript:** In line 47 of the revised manuscript, replaced: "since they have cross sectional areas of ~100 km$^2$"

with "A study by Dickson et al. (2009) found that 53 % of the ISSRs they observed were between 100 and 1500 m deep, and the large eddy simulations conducted in Lewellen (2014) had widths of ~50 km in the transversal direction (defined to be along the horizontal plane perpendicular to the flight direction). Furthermore, a single flight was identified as responsible for creating a cirrus cloud with a bounding box width of 130 km (measured from Fig. 12 (c) in the study by Haywood et al. (2009)). Therefore, the largest persistent contrails can reach cross-sections of up to ~100 km$^2$ in the transversal direction, making gridded simulations of sufficient resolution computationally expensive."

Haywood, J. M., Allan, R. P., Bornemann, J., Forster, P. M., Francis, P. N., Milton, S., Rädel, G., Rap, A., Shine, K. P., and Thorpe, R.: A case study of the radiative forcing of persistent contrails evolving into contrail-induced cirrus, J. Geophys. Res.-Atmos., 114, https://doi.org/10.1029/2009JD012650, 2009.

**"Line 47: This sentence is very general and contains little information. Which models were compared to each other? In which paper was the comparison done? What are the main findings?"**

**Author's response:** We agree with this comment and have improved the description of the ways in which the previous study agree and disagree with each other.

**Author's changes in manuscript:** Replaced "The limited comparisons that have already been performed for large eddy simulations indicate disagreement in this regard (Unterstrasser and Gierens, 2010a; Unterstrasser and Gierens, 2010b; Lewellen et al., 2014; Lewellen, 2014)"

with "A study employing full-lifetime large eddy simulations (Lewellen, 2014) compared its findings with a prior similar study (Unterstrasser and Gierens, 2010a – UG10a; Unterstrasser and Gierens, 2010b – UG10b), and found that "some of the inferences given in UG10a and UG10b are not supported by the present study" and "several of the parameter dependencies discussed here were found previously in UG10a and UG10b" (Lewellen, 2014). Specifically, both studies determined that the total extinction (a proxy climate impact metric) increases with the relative humidity, temperature, and initial contrail ice number. However, they found different parameters dominating the changes in total extinction: relative humidity in Unterstrasser and Gierens (2010a and 2010b), and shear in Lewellen et al. (2014) and Lewellen (2014). Since two models of similar complexity found different dominant factors in predicting a proxy for contrail climate impact, this suggests the need for a more comprehensive assessment of the robustness of contrail modelling techniques being used to inform contrail impact mitigation" in line 56 of the revised manuscript.

**"Line 77: I thought Cocip only tracks a Gaussian plume for the ice crystals and the humidity is taken from NWP data. Why is it necessary to have a plume of water vapour concentrations?"**

**Author's response:** This is true; CoCiP does not assume a Gaussian distribution for the water vapor, only the ice mass. However, CoCiP does keep track of the mass of air in the plume while assuming 100% RHi internally, hence keeping track of the water vapor within the contrail implicitly. Appendix F gives more information about why this is relevant for the study. To address the inaccuracy in the CoCiP description, we have removed "water vapor" from line 98 of the revised manuscript, which talks about the Gaussian assumption in CoCiP.

**Author's changes in manuscript:** Removed "water vapor and" from line 98 of the revised manuscript.

**"Line 96: are the bins fixed in radius space or dynamic as in Lewellen 2014?"**

**Author's response:** APCEMM uses ice radius bins with fixed widths in radius space, but with a dynamic modal radius within each bin (Fritz et al., 2020). Specifically, it implements the scheme described by Jacobson (1997). We have added this description to the text.

**Author's changes in manuscript:** Added the following sentence to Section 1.1.2 in line 122 of the revised manuscript: "These bins are fixed in radius space, but the modal radius of each bin is allowed to increase within the bin bounds to accommodate the increase in ice crystal sizes with time."

**"Line 82: 'evaporate' or is it 'vanish/disappear'?"**

**Author's response:** We extend our appreciation for highlighting the misuse of terminology referring to specific physical processes. We meant to say "disappears".

**Author's changes in manuscript:** Change "evaporates" to "disappears" in line 103 of the revised manuscript.

**"Line 114: I am not sure whether the title is appropriate. Could you reformulate it? "Meteorological background scenarios/data"??"**

**Author's response:** We agree with the suggested change and have implemented it.

**Author's changes in manuscript:** Changed the title of section 2.1 from "Weather parametrization" to "Description of the background meteorology"

**"The quantities you define in Eqs. 1 and 2 have been used in previous studies, yet with other names. It would be good to make the connection to those studies. Unterstrasser & Gierens 2010 introduced the total extinction, which is equal to your definition of γ. Since then, total extinction has also been evaluated in the context of GCM contrail simulations (Bier et al, 2017). Moreover, total surface area S in Lewellen 2014 is basically the same as total extinction (except for a constant scaling factor of 2). Your definition of "lifetime optical depth" was introduced as '(life)time-integrated total extinction' in Unterstrasser (2020). I would recommend to stick to one of the names that have been previously introduced to make clearer that all these studies analyse basically the same quantity."**

**Author's response:** We agree with this comment, which concurs with a similar concern by Anonymous Reviewer #1 (RC1-05). We have chosen to use the "total extinction" term from the 2010 papers by Unterstrasser and Gierens.

**Author's changes in manuscript:** We have renamed our "lifetime optical depth" variable with the "time-integrated total extinction" throughout the manuscript.

**"The crystal loss rate is not well-introduced and I stumble across the units. Is the logarithmic derivative of N(t) used?"**

**Author's response:** We are grateful for having received feedback on the clarity of the formulation for the crystal loss rate, which is indeed the logarithmic derivative. We agree that the absence of its formulation in the original version of the manuscript makes it confusing for the reader. We have addressed this in the manuscript.

**Author's changes in manuscript:** Added the following sentence to the end of section 2.2 in line 178 of the revised manuscript "Due to the order of magnitude changes in total ice number throughout the contrail lifetime, we define the ice crystal loss rate, $-\frac{d\log_{10} N}{dt}$ In decades per hour."

**"Around line 190: You analyse dI/dt and d2I/dt2 which serve as conditions in a contrail phase classification. It makes the impression that those conditions can used as classification criterion across different scenarios. I doubt that the signs of these two quantities are universally interpretable as they may depend on many parameters (such as the thickness of the moist layer, vertical air motions and so forth)."**

**Author's response:** This comment raises a valid point regarding the implied applicability of the criteria determining the diffusion sub-regimes. We agree that, under realistic meteorology, our definitions may not hold. We would like to clarify, however, that we are not attempting to make a universal classification. Instead, we are trying to understand the behavior of the system. We have added a comment to clarify this in the revised manuscript.

**Author's changes in manuscript:** Replaced: "Using the simplified description of the contrail cross-section, mathematical definitions of the sub-regimes observed in Fig. 5 can be formulated by considering the total ice mass per unit length (I)"

with "Mathematical definitions of the sub-regimes observed in Fig. 5 can be formulated by considering the total ice mass per unit length (I), with the caveat that they are only likely to be valid for contrails simulated in idealized meteorology" in line 220 of the revised manuscript.

**"Section 3.2.1: The study by Bier et al (2017) also analysed what factors limit the contrail lifetime. Matching maximum ISSR lifetimes alone are not a sufficient criterion ensuring that your meteorological background state is representative. The characterization of the sub-regimes is more complicated in scenarios where the background humidity changes over time due vertical air motions and ice mass evolution changes by those 'external' drivers. Hence, your claim of widespread applicability is probably a bit overselling."**

**Author's response:** We agree that the "widespread applicability" claim is not accurate, especially since contrails can disappear through synoptic processes and not through sedimentation. We now limit our claim to focus on long-lived persistent contrails only.

**Author's changes in manuscript:** Replaced "may have widespread applicability" to "may be applicable to some long-lived persistent contrails, likely including some of the contrails that are

responsible for 80 % of the climate impact (Teoh et al., 2024)." in line 273 of the revised manuscript.

**"If I understand Fig. 6a correctly, the x-coordinate for the blue and the according orange data point are the same. Correct?**

**Author's response:** Yes, that is correct. We have added a clarifying statement to the caption of Fig. 6 to address this.

**Author's changes in manuscript:** Added "Each entry in the parity plot corresponds to one simulation." To the caption of Figure 6.

**I understand the information given in the text about which fraction of the contrail lifecycle is unobservable (based on τ<0.1). Basically, Cocip contrails are nearly always observable and 100% of their lifecycles belong to the fallstreak regime. Due to these rather peculiar values, the panel b is difficult to understand. First of all, the legends in the two panels say 'fallstreak only' and 'fall streak'. Is this the same criterion?**

**Author's response:** We appreciate that the original version of panel b in Fig. 6 was confusing. As pointed out in comment RC3-02, CoCiP does not really have a fallstreak. We have removed the purple dots in Fig. 6(b) since they were distracting from the intended message.

**Author's changes in manuscript:** Removed the purple dots in Fig. 6(b).

**In the text you mention that on an aggregate level, 92% of ΓAPCEMM comes from post-fall streak regime. It is not explained how you derive this number. Is this the ratio of the orange and blue slope in Fig. 6a? Are all data points equally weighted in the averaging? Do you take the average over the ratios ΓAPCEMM, fallstreak / ΓAPCEMM all ?Or do you sum up over ΓAPCEMM, fallstreak and ΓAPCEMM all separately and then compute the ratio of the two sums? Similarly, I miss information about how the values 35% and 15% (in lines 248 and 249) are computed. Are these the mean values of the orange data points in x and y direction in Fig. 6.2?"**

**Author's response:** We agree with the critique about the clarity of our statistical analysis. The 92% is a ratio of sums, calculated by adding all integrated total extinction after the unrestrained sub-regime across all simulations. This number is divided by the sum of the integrated total extinction across all simulations. No weighing is performed. The 35 % is a similar ratio of sums,

with the threshold being the observability point. We have clarified how these and similar quantities are calculated in the manuscript.

**Author's changes in manuscript:** Amended the text in Section 4.1.1 for clarity: "Figure 6(a) compares the time-integrated total extinction from CoCiP and APCEMM when considering all contrail lifetime (orange) and when only considering the unrestrained sub-regime (purple).

The CoCiP and APCEMM simulations disagree regardless of whether the entire lifetime or the unrestrained sub-regime are considered in isolation. When only the APCEMM unrestrained sub-regime is considered, CoCiP simulations have time-integrated total extinction values 3.3 times larger than those from the corresponding sub-regime in APCEMM (given by the reciprocal of the slope of the purple dashed line in Fig. 6(a)). The case in which all sub-regimes are considered lies above the parity line, with APCEMM simulations having a time-integrated total extinction 3.8 times that of CoCiP (given by the slope of the orange dotted line in Fig. 6(a)).

The relationship between the proportion of the time-integrated total extinction in the unrestrained sub-regime and unobservable regions is displayed in Fig. 6(b). Considering the following sums across all 14 unique simulations:

$$\sigma = \frac{\sum_{\text{all cases}}(\hat{E}_{\text{model}}(t=t*))}{\sum_{\text{all cases}}(\hat{E}_{\text{model}}(t=24\text{ h}))}, \tag{5}$$

where t* is a chosen integration threshold, we find that 92 % of APCEMM time-integrated total extinction is produced after the unrestrained sub-regime, and 38 % is produced when the contrail is unobservable. In contrast, across all simulations CoCiP produces none of its time-integrated total extinction beyond the unrestrained, and 17 % beyond the observability threshold." This starts at line 281 of the revised manuscript.

**"section 4.2.2: You compare APCEMM and Cocip sensitivities with those found in Lewellen 2014. Unterstrasser & Gierens 210a,b also studied contrails in scenarios with constant RHi and analysed the sensitivities to most of the parameters listed in your table 2. Hence, it would help including the findings from these studies in your discussion."**

**Author's response:** We recognize that including the Unterstrasser and Gierens paper will improve the strength of the conclusions and the scientific quality of this manuscript. We accept the suggestions with appreciation.

**Author's changes in manuscript:** Added the following sentence to line 369 of the revised manuscript "Similarly, Fig. 4 and Fig. 6 in Unterstrasser and Gierens (2010a) also confirm that increasing the layer RHi increases the ice mass and the total ice crystal count respectively.".

Replaced the following sentence in line 395 of the revised manuscript:

"Lewellen (2014) find qualitatively similar results to APCEMM: increasing the shear leads to higher ice masses earlier and lower lifetimes" with "Other studies find qualitatively similar results to APCEMM: increasing the shear leads to higher ice masses earlier (Lewellen, 2014 and Unterstrasser and Gierens, 2010a) and lower lifetimes (Lewellen, 2014)."

**"Abstract, first line: what does 'optimized' imply?"**

**Author's response:** We agree that the implications of the word 'optimized' are not clear in the abstract and throughout the rest of the manuscript. By "optimized" strategy, we mean strategies which aim to avoid only the most warming contrails. Such strategies are "optimized" because they reduce the fuel burn penalty relative to more aggressive strategies. Nevertheless, the implication is unclear, so we have replaced the words "optimized" where appropriate in the manuscript.

**Author's changes in manuscript:** Removed the words "optimized" and "optimized strategies" when talking about avoidance strategies. Substituted "optimizes strategies" with "strategies using contrail models to select the specific contrails to avoid" in line 21 of the revised manuscript. Replaced "strategies involving optimized avoidance" with "strategies involving the prioritization of specific contrails by warming" in line 32 of the revised manuscript.

**"Line 56: sublimation is a specific physical process, whereas formation and persistence are more general terms. Moreover, contrails can disappear through other physical processes. Replace 'sublimation' by 'demise'?"**

**Author's response:** We agree that the words "sublimation" and "evaporation" (as well as their conjugated forms) were often misused in the manuscript. We have corrected the oversight.

**Author's changes in manuscript:** changed "sublimation", "evaporation" to "demise" throughout the manuscript. The conjugated forms "sublimates" or "evaporates", have been changed to "disappears".

**"Fig.2: Would it be possible to use white as colour for the zero IWC bin, which would help to better identify the borders of the contrail?"**

**Author's response:** We appreciate that distinguishing the low values of IWC from the background is difficult. We have replotted Fig. 2 with a mask that cuts off IWC beneath $10^{-7}$ kg kg$^{-1}$:

[Figure]

**Author's changes in manuscript:** Changed Fig.2 to distinguish between the contrail and the background.

**"In Eq.3, should the index be 3 (and not 'n')?"**

**Author's response:** We are grateful for this correction and have implemented it.**Author's changes in manuscript:** Changed the subscript of the first Gamma in Eq.3 from "n" to "3".

**"Line 199: "During the fall streak"?? and "fastest center of mass fall rate" (a rate is not fast, it is large)"**

**Author's response:** An omission of the word "sub-regime" and a correction in the description of the center of mass fall rate are correctly highlighted in this comment. We agree with the point raised, and we have clarified the start of line 199.

**Author's changes in manuscript:** Replaced "During the fallstreak" with "During the unrestrained sub-regime" and "fastest" with "largest" in line 233 of the revised manuscript.

**"Line 214: I am not sure what local optical depth means."**

**Author's response:** We agree that "local" optical depth is not properly defined in the text. There are three instances of its use in the manuscript. In each of these instances, we either meant to say "vertical" or "average vertical" optical depth. We have changed this as required.

To be more specific, the contrail optical depth is calculated as follows:

- APCEMM: Each grid cell has an optical depth, which is related to the amount of ice water in the grid cell. The vertical optical depth (a function of width) in APCEMM is the sum of all grid cell optical depths for a given column.
    - The average optical depth is the vertical optical depth integrated along the width coordinate and divided by the width of the contrail.
- CoCiP: The optical depth is calculated for the whole contrail at once (Eqs. 58 – 61 in Schumann (2012)). In a sense, it is analogous to the average vertical optical depth calculated in APCEMM.

**Author's changes in manuscript:** Replaced "local optical depth" with "vertical optical depth" or "average vertical optical depth" as appropriate throughout the manuscript.

**"Line 215: 'is produced at times where …'. Better use time instead of point to make clear it is about time and not space. Moreover, I would prefer to use plural to make clear you consider a time span over which the contrail is not detectable."**

**Author's response:** We agree fully and have adopted this suggestion.

**Author's changes in manuscript:** Changed "is produced at a point where" to "is produced at times when" in line 250 of the revised manuscript.

**"Caption Fig.6: 'unobservable'"**

**Author's response:** We recognize this oversight and have fixed it in the revised manuscript.

**Author's changes in manuscript:** Changed the misspelled word in the caption of Fig. 6 to "unobservable".

**"Line 262: 'at the end of the APCEMM fallstreak regime(?)'; 'shear does not increase the contrail width'. It is true that shear increases the contrail width. But here you want to say that a larger shear value leads to a larger contrail width."**

**Author's response:** We agree and have rephrased the analysis to be more accurate. Furthermore, we omit the word "sub-regime" after the "APCEMM fallstreak" in the original manuscript.

**Author's changes in manuscript:** We change line 309 of the revised manuscript from "shear increases the contrail width by 143 % and the ice mass by 58 %, leading to a 36 % increase in lifetime optical depth" to "higher shear leads to an increase of the contrail width by 143 % and an increase of the ice mass by 58 %, leading to a 36 % increase of the time-integrated total extinction".

**"Line 337: 'once the fallstreak ends': in time or space?"**

**Author's response:** This refers to the point in time. We have modified the sentence in an attempt to improve the clarity.

**Author's changes in manuscript:** Replaced the sentence "It is also helpful to consider the effect that wind shear has on a contrail once the fallstreak ends" with "It is also helpful to consider the effect that wind shear has on a contrail after the time when the unrestrained sub-regime ends" in line 408 of the revised manuscript.

**"Line 345 I believe it should be 'Contrail avoidance strategies that' because the following clause is restrictive. Same in line 385: 'which' -> 'that'."**

**Author's response:** We agree and have changed the language accordingly.

**Author's changes in manuscript:** Replaced the sentence "Contrail avoidance which does not attempt" in line 416 of the revised manuscript with "Contrail avoidance strategies that do not attempt". Other replacements of "which" with "that" throughout the text are made as appropriate

**"Line 348: and also lifetime-integrated radiative effects?"**

**Author's response:** Although we recognize the added value that mentioning radiative effects would bring to the manuscript, we have decided not to include a specific reference to them after very careful consideration. This is because we have not performed any radiative transfer calculations. We see the primary purpose of this study to investigate the plume models without any supplements. We do admit, however, that this scope is limited given the global scale of the contrails problem. Therefore, we have updated the limitations section in the manuscript to recommend further studies running comparisons which include both radiative transfer calculations and a global analysis.

**Author's changes in manuscript:** Added this sentence to line 473 of the revised manuscript: "Finally, to determine the full extent of the climate implications of the comparison, we encourage future studies to include radiative transfer calculations on a set of contrail simulations around the globe."

---

## Author Comment (AC3)

**Responses to Community Comments (from Sina Hofer and Klaus Gierens)**

**"In putting their results and model assumptions into perspective, the authors feel the need to argue that ISSRs can last many hours. In order to support this, the authors write:**

**"This is further supported by a recent preprint (Hofer and Gierens, 2024) which analyzed a larger ECMWF dataset and found that contrail lifetime is most commonly limited by sedimentation, as opposed to advection of the contrail out of the ISSRs."**

**Here, we find an error and a misinterpretation. The error is: we have used data from the ICON model of the German Weather Service, not ECMWF data.**

**The misinterpretation is: Although the movement of ISSRs is often aligned to the wind, this does not imply that contrail lifetimes are most commonly limited by sedimentation. To the contrary, in another recent preprint (Hofer and Gierens 2025) we show that contrail termination by sedimentation and contrail termination by synoptic processes (contrails leaving the ISSR with the wind and large-scale subsidence turning super- into subsaturation) have similar time-scales of a few hours. It is difficult to say in advance which time-scale is shorter."**

**Author's response:** This comment raises a misunderstanding and an error in Section 3.2.1. We have corrected the error and the misconception in the revised manuscript.

**Author's changes in manuscript:** Replaced "which analyzed a larger ECMWF dataset" with "which analyzed the ICON dataset" in Section 3.2.1.

Replaced "found that contrail lifetime is most commonly limited by sedimentation, as opposed to advection of the contrail out of the ISSRs" with "found that contrail lifetime is most commonly limited by sedimentation and synoptic processes such as advection of contrails out of the ISSRs".

Added the statement "Further, a second preprint by Hofer and Gierens (2025) found that the sedimentation and synoptic timescales are both in the order of a few hours."

We also added a comment on the new preprint to Section 4.1.3: "To understand the sensitivity of our findings to the contrail lifetime, we define the global model difference ($\underline{\delta}$) as the sum across all simulations of the APCEMM integrated total extinction minus the CoCiP integrated total extinction (at each timestep):

$$\delta(t) = \sum_{\text{all cases}}\left(\widehat{E}_{\text{APCEMM}}(t) - \widehat{E}_{\text{CoCiP}}(t)\right), \tag{6}$$

$$\widehat{\delta}(t) = \frac{\delta(t)}{\delta(t = 24\,\text{h})}, \tag{7}$$

where $\widehat{\delta}(t)$ is the normalized global model difference. The variable t in Eqs. 5 and 6, is the upper limit of integration in Eq. 2.

Figure 10 shows how $\widehat{\delta}(t)$ varies as a function of time. We hence find that 90 % of the global model difference is produced within 12 hours from formation. For more evidence-based contrail lifetime estimates, we take 4 h and 8 h from a recent preprint by Hofer and Gierens (2025). The

proportion of the total model difference reached by 4 h and 8 h are 27 % and 72 % respectively. These results indicate a large sensitivity in our findings to the lifetime of typical contrails. However, they also indicate that our findings are particularly relevant to those 6–7 % of contrails that persist beyond 8 h (Gierens and Vazquez-Navarro, 2018). Such contrails are also likely to be the greatest contributors to aviation warming on an individual basis and are hence important for contrail avoidance."

**A similar statement is found in the conclusion: "the predominant mechanism for contrail evaporation is through sedimentation, as opposed to advection (Hofer and Gierens, 2024; Irvine et al., 2024)." We do not know Irvine et al. wrote (by the way, this was 2014, not 2024), but it is not our statement.**

**Author's response:** This comment raises an oversight and a misunderstanding in Section 6. Other than the typo in the citation to Irvine et al (2014), upon further review, we believe that the way the citation is used is incorrect since Irvine et al (2014) talked about ice supersaturated regions, and not contrails. We have also addressed the misinterpretation from the comment CC-01 in the conclusion.

**Author's changes in manuscript:** We removed the citation to Irvine et al (2014) in line 375 of the original manuscript, and modified the erroneous statement to reflect the findings from the most recent preprint by Hofer and Gierens in the paragraph starting at line 448 of the revised manuscript: "However, tropospheric ice supersaturated regions are generally sufficiently large that contrail demise occurs through sedimentation, synoptic processes, or both at similar timescales (Hofer and Gierens, 2024; Hofer and Gierens, 2025). Furthermore, 72 % of the model disagreement on the time-integrated total extinction can be attributed to the first 8 hours of the simulations (see Sect. 4.1.3). This makes the conceptual findings from this study applicable to those real contrails which persist for long."

**More arguments for long-lasting ISSRs can be found in case studies by Bakan et al. and Spichtinger et al., if needed.**

**Author's response:** We are grateful for having received specific pointers to references that support our work. However, with the new preprint by Hofer and Gierens and our sensitivity analysis to lifetime, there is no need to further substantiate the point we were trying to make.

**Author's changes in manuscript:** No changes to the manuscript.

**Statistical arguments about the unobservable fraction of contrail lifetime can be found in a paper be Gierens and Vazquez-Navarro (2018). You might want to check whether your results agree with those from the statistical arguments.**

**Author's response:** We greatly appreciate this suggestion, and agree. We now cite Gierens and Vazquez-Navarro (2018) in Section 3.2.1 of the revised manuscript, citing the proportion of contrails with lifetimes over 8 hours.

**Author's changes in manuscript:** Added the following sentences to 3.2.1 (line 270 of the revised manuscript): "This is corroborated by a study estimating the full-lifetime of contrails with statistical methods applied to satellite observations (Gierens and Vazquez-Navarro, 2018). Interpolating Fig. 8 from Gierens and Vazquez-Navarro (2018), it can be estimated that the proportion of contrails with lifetimes exceeding 8 h is ~6–7 %."

---

## Author Response (AR2)

**Final Author Reply to the Editor (2nd Round)**

Caleb Akhtar Martínez, Sebastian D. Eastham, and Jerome P. Jarrett

Dear Prof. Martina Krämer,

We are very grateful to you for your consideration of our manuscript, and to the referees for their second round of constructive feedback. These comments have further enhanced our scientific rigor and analysis. Our detailed responses to each point are provided below.

As in our previous response, we are using the following structure: (1) Referee Comment; (2) Response; and (3) Changes in Manuscript. All changes have been marked in the revised version for ease of reference. In this revision, we have addressed the minor comments highlighted by the reviewers, and we do not believe there are any changes that require highlighting.

We hope that the revised manuscript now meets the expectations of the reviewers and ACP, and we respectfully submit it for your consideration.

Sincerely,

Caleb Akhtar Martínez, Sebastian D. Eastham, and Jerome P. Jarrett

**Responses to Anonymous Referee #1**

"L 105 and 106: Is it actually "m depth" or rather "m of flightpath"? The latter would sound more familiar to me."

**Author's response:** We are grateful for this comment and agree that stating "m of flightpath" causes less confusion.

**Author's changes in manuscript:** Changed "m depth" to "m of flightpath" on line 108 of the revised manuscript.

"L 110: I assume, r means the radius of the ice crystals. This should be written."

**Author's response:** Agreed. We have now clarified this in the manuscript.

**Author's changes in manuscript:** Clarified that r refers to the average ice crystal radius in line 112 of the revised manuscript "of order r – the average ice crystal radius".

**Responses to Anonymous Referee #3**

"Fig. 2: what is the cruise altitude in the used coordinate system? Please add this information in the caption or plot."

**Author's response:** The cruise pressure altitude in Fig. 2 is 10000 m. We agree that it would be beneficial to add this in the caption.

**Author's changes in manuscript:** Added "The origin of each panel represents the approximate visual center of the contrail, at a pressure altitude of 9927 m in CoCiP and 9750 m in APCEMM. The cruise altitude is 10000 m." to the caption of Fig. 2.

"Line 105: not sure if ,m depth' is appropriate here. You should better explain that your quantities are given per unit length along flight distance."

Author's response: Agreed. This has been changed in the revised manuscript.

**Author's changes in manuscript:** Changed "m depth" to "m of flightpath" on line 108 of the revised manuscript.

",Weather data' in the green bubble in figure 3 is (still) misleading as your meteorological background conditions are very idealised scenarios and weather data in my opinion implies that you use NWP data or something similar."

**Author's response:** We are grateful for highlighting this oversight and agree that Figure 3 should also be changed to reflect the use of synthetic meteorology.

**Author's changes in manuscript:** Changed the text in the green bubble in Figure 3 from "Weather data" to "Synthetic Meteorology"

"You present a new type of analysis in section 4.1.3, which is in general a nice approach. However, I am not sure how meaningful it is in the present form. In Eq. 6, the error could be zero, when the under/overestimations cancel out by summing over all cases. Would it not make more sense to sum up the absolute values of model differences?"

**Author's response:** We agree with the perspective shared here and we have concluded that the analysis would be clearer to the reader if the absolute value is used in lieu of the signed error.

**Author's changes in manuscript:** Replaced the brackets in Eq. 6 with absolute values. There are no changes in the results because the time-integrated total extinction is greater in APCEMM than in CoCiP (at all times) in all simulations.

"Moreover the reasoning around line 345 is not very rigorous. Contrail lifetimes of 4h and 8h occur in other meteorological scenarios than you studied. Hence it is speculation how model differences would look like in such atmospheric scenarios. Contrail sublimation due to subsidence may occur at different rates in the two models. Hence \delta(4h) of your simulation set does not capture this."

**Author's response:** It is true that the reasoning presented in line 345 is speculative. We agree with the points raised in this comment, and we have adjusted our choice of language to reflect the speculative nature of the analysis.

**Author's changes in manuscript:** Changed "However, they also indicate that our findings are particularly relevant to those 6–7 % of contrails that persist beyond 8 h" to "We also hypothesize that our findings could be particularly relevant to those 6–7 % of contrails that persist beyond 8 h" in lines 348-349 of the revised manuscript.

"Around line 450 and in some prior text parts, you state that the long contrail lifetimes observed in your scenarios may match observations of contrails with the greatest warming to highlight the relevance of your results. Given that your strongly idealised background conditions use a rather academic time-constant RHi I suggest to not overstate the conclusiveness of your comparison. Similar lifetimes do not guarantee that your scenarios are good proxies for strong-contrails scenarios."

**Author's response:** We agree that our outlooks are speculative and have adjusted our language accordingly in Sections 3.2.1, 4.1.3, and the conclusions.

**Author's changes in manuscript:**

In the conclusions, we changed "Since Gierens and Vazquez-Navarro (2018) found that  $\sim$ 6–7 % of contrails persist beyond 8 hours, this makes it likely for the conceptual findings from this study to be applicable to the real contrails with the greatest warming." to

"Since Gierens and Vazquez-Navarro (2018) found that ~6–7 % of contrails persist beyond 8 hours, it is possible that the conceptual findings from this study could be applicable to the some of the real contrails responsible for most of the warming. Nevertheless, further comparisons using more realistic meteorology are needed to validate our findings and to reveal the extent of their applicability." in lines 455 to 458 of the revised manuscript.

In Section 3.2.1, we added the following sentence "Nevertheless, the extent of the applicability of our findings on real contrails will need to be determined through further experimental and computational work" to line 277 of the revised manuscript.

In Section 3.2.1 we also replaced "likely including some of the contrails" to "perhaps including some of the contrails" in line 276 of the revised manuscript.

In Section 4.1.3 we changed "However, they also indicate that our findings are particularly relevant to those 6–7 % of contrails that persist beyond 8 h" to "We also hypothesize that our findings could be particularly relevant to those 6–7 % of contrails that persist beyond 8 h" in lines 348-349 of the revised manuscript.